
# Multiloop functional renormalization group for the two-dimensional Hubbard model: Loop convergence of the response functions

Agnese Tagliavini[1,2], Cornelia Hille[1⋆], Fabian B. Kugler[3],
Sabine Andergassen[1], Alessandro Toschi[2] and Carsten Honerkamp[4,5]

**1** Institut für Theoretische Physik and Center for Quantum Science, Universität Tübingen,
Auf der Morgenstelle 14, 72076 Tübingen, Germany
**2** Institute for Solid State Physics, Vienna University of Technology, 1040 Vienna, Austria
**3** Physics Department, Arnold Sommerfeld Center for Theoretical Physics,
and Center for NanoScience, Ludwig-Maximilians-Universität München,
Theresienstrasse 37, 80333 Munich, Germany
**4** Institute for Theoretical Solid State Physics,
RWTH Aachen University, D-52056 Aachen, Germany
**5** JARA - Fundamentals of Future Information Technology

⋆ cornelia.hille@uni-tuebingen.de

## Abstract

We present a functional renormalization group (fRG) study of the two dimensional Hubbard model, performed with an algorithmic implementation which lifts some of the common approximations made in fRG calculations. In particular, in our fRG flow; (i) we take explicitly into account the momentum and the frequency dependence of the vertex functions; (ii) we include the feedback effect of the self-energy; (iii) we implement the recently introduced multiloop extension which allows us to sum up *all* the diagrams of the parquet approximation with their exact weight. Due to its iterative structure based on successive one-loop computations, the loop convergence of the fRG results can be obtained with an affordable numerical effort. In particular, focusing on the analysis of the physical response functions, we show that the results become *independent* from the chosen cutoff scheme and from the way the fRG susceptibilities are computed, i.e., either through flowing couplings to external fields, or through a "post-processing" contraction of the interaction vertex at the end of the flow. The presented substantial refinement of fRG-based computation schemes paves a promising route towards future quantitative fRG analyses of more challenging systems and/or parameter regimes.

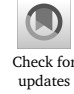
# 1  Introduction

Over the last two decades, functional renormalization group (fRG) methods have been broadly used for analyzing two-dimensional (2D) lattice electron systems (for reviews, see Refs. [1,2]). The main advantage of the fRG lies in the exploration of the leading low-energy correlations and instabilities towards long-range ordered states, similar to what has been investigated earlier for one-dimensional systems [3–5]. However, in one dimension, other methods like Bethe-Ansatz, bosonization [6,7] and DMRG [8] exist, which are for certain aspects more controlled. Hence, assessing the precision of RG methods in one-dimensional systems was not really in the

foreground. The situation evidently changes for two- and three-dimensional systems, where the specific simplifications associated to the peculiar one-dimensional geometry are not applicable. At the same time, spatial correlations in 2D are strong enough to induce qualitative corrections [9,10] with respect to another class of rigorous many-body approaches, such as the Dynamical Mean-Field Theory (DMFT) [11–13] which allows one to include all purely local dynamical correlations.

In fact, due to the intrinsic complexity of the many-electron problem in 2D, the development of unbiased quantitative methods applicable to a wide energy range from electronic structures on the scale of a few eV down to, e.g., ground state ordering in the (sub-)meV region is still on the wishlist. This goal has motivated, in the last decade, the development of several algorithmic schemes for treating electronic correlations in 2D from different perspectives [1, 14, 15]. In this context, the fRG has already unveiled quite promising features: The fRG has the potential of resolving band structures and Fermi surface details *and* to treat competing orders on low energy scales in a rather unbiased way, since it does not require preliminary assumptions about dominating scattering channels. Recent applications range from studies of cuprate high-$T_c$ superconductors [16–19] over iron superconductors [2,20] to few-layer graphene systems [21, 22], to cite a few.

We also note that, while the current applicability of the fRG is generally restricted to the weak to intermediate coupling regimes, its combination [23, 24] with the DMFT might allow one, in the future, to access much more strongly correlated parameter regions, including the ones in proximity of the Mott-Hubbard metal-insulator transition. This is achieved by constructing a fRG flow starting from the DMFT solution of the considered lattice problem to the exact solution, i.e., in practice, using the DMFT to determine the initial conditions for the fRG flow [23]. Similarly to other diagrammatic extensions [15] of DMFT, such as the Dynamical Vertex Approximation (DΓA) [25] or the Dual Fermion [26] approach, one might work either with the physical degrees of freedom (as in the so-called DMF$^2$RG [23]) or in the space of auxiliary (dual) fermions [27], introduced by means of a suitable [15, 26] Hubbard-Stratonovich transformation.

Yet, what is hitherto missing is a thorough analysis of the *quantitative* reliability of the fRG for a well-defined test case. More precisely this would require to clarify how much the fRG results, going beyond the correct estimation of general physical trends, depend on the approximations inherent in the used fRG scheme. This study within the fRG would then also provide a solid basis for future comparisons with other numerical techniques.

The mentioned approximations can be grouped in three categories:

(i) *Momentum/frequency discretization*: As the fRG algorithm typically exploits the flow of vertex functions that depend continuously on multiple momenta and frequencies, various approximations are performed to mitigate numerical and memory costs. Early on, $N$-patch discretizations of the momentum dependencies through the Brillouin zone were used. Later, it was noticed that channel-decompositions in conjunction with form factor expansions [28–30] lead to physically appealing approximations featuring advantageous momentum resolution and numerical performance [31]. Clever prescriptions for the treatment of the high-frequency tails of the vertex function have been devised [32–34] which are also used in this work.

(ii) *Self-energy feedback*: In many applications of the fRG the self-energy and its feedback on the flow of the $n$-particle ($n > 1$) vertex functions has *not* been accounted for. While there are arguments that the self-energy may be important mainly when the interactions are close to a flow to strong coupling (see Appendix in Ref. [35]), more quantitative results should overcome this deficit. In fact, neglecting the self-energy feedback was mainly motivated by the disregarded frequency dependence of the interactions in earlier fRG studies: Within a static treatment the self-energy lacks the effects of quasiparticle degradation, so that its inclusion became less important. Within the current frequency-dependent fRG treatments, the self-energy

feedback can be included in a meaningful way. A number of works have already investigated the self-energy effects in the flows to strong coupling in Hubbard-type models [29, 36–44], mainly exploring the quantitative effects, besides signatures of pseudogap openings [39, 40] and non-Fermi liquid behavior [29] in particular cases.

(iii) *Truncation of the flow equation hierarchy*: Finally, one should also consider the truncation of the hierarchy of flow equations for the $n$-point one-particle irreducible (1PI) vertex functions. This is usually done at "level-II" as defined in Ref. [1], also referred to as one-loop ($1\ell$) approximation, i.e., the 1PI six-point vertex is set to zero. Due to this truncation, the final result of an fRG flow might depend – to a certain degree – on the cutoff scheme adopted for the calculation.

In this perspective, it was noticed by Katanin [45] that replacing the so-called single-scale propagator in the loops on the r.h.s. of the flow equation for the four-point vertex by a scale-derivative of the full Green's function allows this scheme to become equivalent to one-particle self-consistent (a.k.a. mean-field) theories in reduced models, and then to go beyond such self-consistent approximations in more general models. Another significant comparison can be made with the parquet-based approaches [46, 47], such as the parquet approximation (PA) [33, 34, 48–50]. The latter represents the "lowest order" solution of the parquet equations, where the two-particle irreducible vertex is approximated by the bare interaction. In fact, although the diagrams summed in the $1\ell$ truncation of the fRG are topologically the same as in the PA, the way the single contributions are generated during the flow leads in general to differences with respect to the PA [34, 51]. This is due to some internal-line combinations, e.g., in particle-hole corrections to the particle-particle channel, which are suppressed by the cutoff functions attached to the propagators and not fully reconstructed during the flow because of the truncation. A quantitative analysis of this effect has been performed for the single impurity Anderson model in Ref. [34]. These differences are absent for single-channel summations (e.g. RPA), but could lead to more pronounced quantitative errors in presence of channel coupling, e.g., in the generation of superconducting pairing through spin fluctuations. Furthermore, while the Mermin-Wagner theorem is fulfilled within the PA [52], it is typically violated by $1\ell$ fRG calculations. First steps to remedy this shortcoming were undertaken in various works [43, 53, 54], but only recently a comprehensive path of how the PA contributions can be recovered in full extent was presented within the multiloop extension of the fRG (mfRG) [55–57]. The mfRG flow equations incorporate all contributions of the six-point vertex that complement the derivative of diagrams already part of the $1\ell$ flow, as organized by their loop structure. A key insight in this approach is that the higher-loop contributions can be generated by computing $1\ell$ flows for scale-differentiated vertices, with an effort growing only linearly with the loop order that is fully kept. The multiloop corrections stabilize the flow by enabling full screening of competing two-particle channels, ultimately recovering the self-consistent structure of the PA. As the PA corresponds to a well-defined subset of diagrams, a converged mfRG flow able to reproduce the PA is by construction independent of the adopted cutoff.

In this paper, we present a fRG study of the 2D Hubbard model performed with an algorithm combining the most recent progress on all three approximation levels. We use (i) the so-called "truncated unity" fRG [31] (TUfRG) formalism to describe the momentum dependence of the vertex and, in addition, keep the full frequency dependence as a function of three independent frequencies. Differently from the approach adopted in Ref. [44], we employ a refined scheme to treat the high-frequency asymptotics [34] that allows us to reduce the numerical effort considerably. Within this scheme, we can consistently include (ii) the (frequency-dependent) self-energy feedback in our fRG flow equations. Finally, we present (iii) first data for the 2D Hubbard model computed with the multiloop extension proposed by Kugler and von Delft [55]. In this context, we have also generalized the multiloop formalism to compute the flow of the response functions, and illustrated the loop convergence of the fRG

results for the 2D Hubbard model. In particular, we show that including up to 8 loops in the fRG flow yields a clear convergence of the data with the loop order and the final results are independent of the cutoff. This represents an important check and illustrates that fRG flows can be brought in quantitative control for 2D problems. Finally, our multiloop analysis of the response functions demonstrates that the two different ways to compute susceptibilities in the fRG, either by tracking the renormalization group flow of the couplings to external fields [1] or by contracting the final interaction vertex (see, e.g., Ref. 23), converge to the same value with increasing loop order. This confirms that the output of this improved fRG scheme can indeed be trusted on a quantitative level.

The paper is organized as follows: The formalism and theory of the linear response functions and their computation by mfRG flow equations are introduced in Section 2. In Section 3 we present the actual implementation scheme for the full momentum- and frequency-dependent fRG. In Section 4 we show the results for the 2D Hubbard model, with a detailed analysis of the effects of the different approximation levels and in particular of the convergence with the loop order. A conclusion and outlook is provided in Section 5.

## 2 Theory and formalism

### 2.1 Definitions and formalism

In this section we provide the definitions of the linear response functions to an external field, before describing their computation with the fRG. We focus on correlation functions of fermionic bilinears. In particular, in a time-space translational-invariant system, we consider the charge (density) and spin (magnetic) bilinears, both charge invariant,

$$\rho_{\mathrm{d}}^{n}(q) = \sum_{\sigma} \int dp \, \bar{\psi}_{\sigma}(p) f_{n}(p,q) \psi_{\sigma}(p+q) \,, \tag{1a}$$

$$\rho_{\mathrm{m}}^{n}(q) = \sum_{\sigma} (-1)^{\sigma} \int dp \, \bar{\psi}_{\sigma}(p) f_{n}(p,q) \psi_{\sigma}(p+q) \,, \tag{1b}$$

and the non-charge invariant pairing (superconducting) bilinears

$$\rho_{\mathrm{sc}}^{n}(q) = \int dp \, \psi_{\downarrow}(q-p) f_{n}^{*}(p,q) \psi_{\uparrow}(p) \,, \tag{2a}$$

$$\rho_{\mathrm{sc}}^{n*}(q) = \int dp \, \bar{\psi}_{\uparrow}(p) f_{n}(p,q) \bar{\psi}_{\downarrow}(q-p) \,, \tag{2b}$$

where $\psi$ and $\bar{\psi}$ represent the Grassman variables and $p$ ($q$) a fermionic (bosonic) quadri-momentum $p = \{i\nu_{o}, \mathbf{p}\}$ ($q = \{i\omega_{l}, \mathbf{q}\}$). The integral includes a summation over the Matsubara frequencies ($i\nu_{o}$), normalized by the inverse temperature $\beta$, and an integral over the first Brillouine Zone normalized by its volume $\mathcal{V}_{\mathrm{BZ}}$. The function $f_{n}(p,q)$ determines the momentum and frequency structure of the bilinears in the different physical channels. In the present case we restrict ourselves to a static external source field, such that the function $f_{n}(p,q) = f_{n}(\mathbf{p})$ acquires only a momentum dependence, whose structure is specified by the subscript $n$ and explicitly shown in Table 1 (in the present work we will mostly focus on the $s$- as well as $d$-wave momentum structure). Note that, when using a different frequency-momentum notation, centered in the center of mass of the scattering process (see "symmetrized" notation in Appendix A), one should account for an additional shift of the momentum dependence $\mathbf{p}$ by means of the momentum transfer $\mathbf{q}$.

After a reshift of the operators in Eq. (1) with respect to their average value $\rho^n_{\text{d/m}} \to \rho^n_{\text{d/m}} - \langle \rho^n_{\text{d/m}} \rangle$, we can now define the correlation functions of these bilinears in the three channels

$$\chi^{nn'}_{\text{d/m}}(q) = \frac{1}{2} \langle \rho^n_{\text{d/m}}(q) \rho^{n'*}_{\text{d/m}}(q) \rangle \tag{3a}$$

$$\chi^{nn'}_{\text{sc}}(q) = \langle \rho^n_{\text{sc}}(q) \rho^{n'*}_{\text{sc}}(q) \rangle. \tag{3b}$$

In linear response theory, these correlation functions correspond to the physical susceptibilities in the corresponding channels. Divergences in $\chi^{nn'}_{\eta}(q)$, with $\eta = \{\text{d}, \text{m}, \text{sc}\}$, indicate spontaneous ordering tendencies or instabilities of the system. The above definition encodes not only the real-space pattern or wavevector for which the system starts ordering, but also the symmetry of the order parameter associated to the instability. In the 2D Hubbard model study presented here (see Section 4) we detect various response functions growing considerably towards low $T$, such as the spin-density wave (SDW) response, characterized by the isotropic $s$-wave magnetic susceptibility at $\mathbf{q} = (\pi, \pi)$, as well as $s$- and $d$-wave pairing response functions at $\mathbf{q} = (0,0)$ and Pomeranchuk instabilities [58]. Inserting Eq. (1) or Eq. (2) into Eq. (3), the susceptibilities appear as two-particle Green's functions. In particular, they can be determined from the two-particle vertex $\gamma_4$ by

$$\chi^{nn'}_{\text{d/m}}(q) = \frac{1}{2} \sum_{\sigma\sigma'} \int dp\, dp'\, f_n(\mathbf{p}) f^*_{n'}(\mathbf{p}') \boldsymbol{\sigma}^{0/3}_{\sigma\sigma} \boldsymbol{\sigma}^{0/3}_{\sigma'\sigma'} \big[ \Pi_{\text{d/m};\sigma,\sigma'}(q,p,p') +$$
$$\Pi_{\text{d/m};\sigma\sigma}(q,p,p) \gamma_{4;\sigma\sigma\sigma'\sigma'}(p,p+q,p'+q,p') \Pi_{\text{d/m};\sigma'\sigma'}(q,p',p') \big] \tag{4a}$$

$$\chi^{nn'}_{\text{sc}}(q) = \int dp\, dp'\, f_n(\mathbf{p}) f^*_{n'}(\mathbf{p}') \big[ \Pi_{\text{sc};\uparrow\downarrow}(q,p,p') +$$
$$\Pi_{\text{sc};\uparrow\downarrow}(q,p,p) \gamma_{4;\uparrow\uparrow\downarrow\downarrow}(p,p',q-p,q-p') \Pi_{\text{sc};\uparrow\downarrow}(q,p',p') \big], \tag{4b}$$

where $\boldsymbol{\sigma}^{0/3}$ represent the Pauli matrices ($\boldsymbol{\sigma}^0 = \mathbb{1}$) and we made use of the spin conservation. Eqs. (4) can be considerably simplified by making use of the SU(2) symmetry. The "bare bubbles" $\Pi_\eta$ appearing in (4) read

$$\Pi_{\text{d/m};\sigma\sigma'}(q,p,p') = -\beta \mathcal{V}_{\text{BZ}} \delta_{\sigma,\sigma'} \delta_{p,p'} G_\sigma(p) G_\sigma(p+q), \tag{5a}$$

$$\Pi_{\text{sc};\uparrow\downarrow}(q,p,p') = \beta \mathcal{V}_{\text{BZ}} \delta_{p,p'} G_\uparrow(p) G_\downarrow(q-p). \tag{5b}$$

By exploiting the SU(2) symmetry,

$$G_\sigma(p) = G_{\bar\sigma}(p) = G(p), \tag{6}$$

we can drop the spin dependencies for the bare bubbles. In presence of the above symmetries, we can introduce the following definitions for (spin-independent) channels of the two-particle vertex

$$\gamma_{4,\text{d}}(q,p,p') = \frac{1}{2} \sum_{\sigma,\sigma'} \gamma_{4;\sigma\sigma\sigma'\sigma'}(p,p+q,p'+q,p') \tag{7a}$$

$$\gamma_{4,\text{m}}(q,p,p') = \frac{1}{2} \sum_{\sigma,\sigma'} (-1)^{\epsilon_{\sigma\sigma'}} \gamma_{4;\sigma\sigma\sigma'\sigma'}(p,p+q,p'+q,p') \tag{7b}$$

$$\gamma_{4,\text{sc}}(q,p,p') = \gamma_{4;\uparrow\uparrow\downarrow\downarrow}(p,p',q-p,q-p'), \tag{7c}$$

with $\epsilon$ the Levi-Civita symbol. The resulting spin-independent expression of the physical susceptibilities reads

$$\chi^{nn'}_{\eta}(q) = \int dp\, dp'\, f_n(\mathbf{p}) f^*_{n'}(\mathbf{p}') \big[ \Pi_\eta(q,p,p') + \Pi_\eta(q,p,p) \gamma_{4,\eta}(q,p,p') \Pi_\eta(q,p',p') \big]. \tag{8}$$

We conclude this section by recalling the definition of the so-called fermion-boson vertex [59], which, for the considered symmetries, reads

$$\gamma^n_{3,d/m;\sigma\sigma}(q,p) = \Pi^{-1}_{d/m}(q,p,p)\boldsymbol{\sigma}^{0/3}_{\sigma\sigma}\langle\bar{\psi}_\sigma(p)\psi_\sigma(p+q)\rho^{n*}_{d/m}(q)\rangle \tag{9a}$$

$$\gamma^n_{3,sc;\downarrow\uparrow}(q,p) = \Pi^{-1}_{sc}(q,p,p)\langle\psi_\downarrow(p)\psi_\uparrow(q-p)\rho^{n*}_{sc}(q)\rangle \,. \tag{9b}$$

Similarly to the susceptibility, one can rewrite Eqs. (9a) and (9b) in a form where the two-particle vertex $\gamma_{4,\eta}$ appears explicitly

$$\gamma^n_{3,\eta}(q,p) = f_n(\mathbf{p}) + \int dp' f_n(\mathbf{p}')\gamma_{4,\eta}(q,p,p')\Pi_\eta(q,p',p') \,, \tag{10}$$

where, because of the SU(2) symmetry, we dropped the spin dependence of the fermion-boson vertices

$$\gamma^n_{3,d/m;\sigma\sigma} = \gamma^n_{3,d/m;\bar{\sigma}\bar{\sigma}} = \gamma^n_{3,d/m} \tag{11a}$$

$$\gamma^n_{3,sc;\downarrow\uparrow} = \gamma^n_{3,sc;\uparrow\downarrow} = \gamma^n_{3,sc} \,. \tag{11b}$$

## 2.2 Flow equations for the response functions

In this section we derive the mfRG [55] flow equations of the response functions and discuss the improvement with respect to the $1\ell$ version [1]. Note that one can also provide a formal analytical derivation of these flow equations [57]. In the following we provide the main steps of the derivation in the 1PI formulation [1,60] (see also Ref. [58] for the Wick-ordered formulation), for the details we refer to Appendix B. Following the review of Metzner et al. [1], we introduce the coupling of the density operators in Eqs. (1) and (2), shifted with respect to their average values, i.e. $\rho^n_\eta \to \rho^n_\eta - \langle\rho^n_\eta\rangle$), to the external field $J_\eta$ by defining the following scalar product

$$(J^n_{d/m},\rho^n_{d/m}) = \int dq J^n_{d/m}(q)\rho^n_{d/m}(q) \,, \tag{12a}$$

$$(J^n_{sc},\rho^{n*}_{sc}) + (J^{n*}_{sc},\rho^n_{sc}) = \int dq\big[J^n_{sc}(q)\rho^{n*}_{sc}(q) + J^{n*}_{sc}(q)\rho^n_{sc}(q)\big] \,. \tag{12b}$$

We note that, although $J^n_\eta$ appears as a functional dependence in our derivation, it is not an integration variable since our system is fully fermionic (for an fRG formulation of coupled fermion-boson systems, see Refs. [1,61–63]).

By expanding the scale-dependent effective action $\Gamma^\Lambda$ in powers of the fermionic fields, as well as of the external bosonic source field, we obtain

$$\Gamma^\Lambda[J_\eta,\bar{\psi},\psi] = \Gamma^\Lambda[\bar{\psi},\psi] + \sum_\eta\sum_{y_1,y_2}\frac{\partial^{(2)}\Gamma^\Lambda[J_\eta,\bar{\psi},\psi]}{\partial J_\eta(y_1)\partial J^*_\eta(y_2)}\bigg|_{\substack{\psi=\bar{\psi}=0\\J=0}}J_\eta(y_1)J^*_\eta(y_2)-$$

$$\sum_{\eta'=d,m}\sum_{y,x,x'}\frac{\partial^{(3)}\Gamma^\Lambda[J_\eta,\bar{\psi},\psi]}{\partial J_{\eta'}(y)\partial\bar{\psi}(x')\psi(x)}\bigg|_{\substack{\psi=\bar{\psi}=0\\J=0}}J_{\eta'}(y)\bar{\psi}(x')\psi(x)-$$

$$\sum_{y,x,x'}\frac{\partial^{(3)}\Gamma^\Lambda[J_\eta,\bar{\psi},\psi]}{\partial J_{sc}(y)\partial\bar{\psi}(x')\partial\bar{\psi}(x)}\bigg|_{\substack{\psi=\bar{\psi}=0\\J=0}}J_{sc}(y)\bar{\psi}(x')\bar{\psi}(x) + ... \tag{13}$$

Note that the index $x = \{\sigma,k\}$ combines the spin index $\sigma$ and the fermionic quadrivector $k = (i\nu_l,\mathbf{k})$ (here we disregard additional quantum dependencies, e.g., orbital), while

$y = \{n, q\}$ refers to the momentum structure of the coupling to the bilinears, $n$, and to the bosonic quadrivector $q = \{i\omega_l, \mathbf{q}\}$. In Eq. (13) the first term on the r.h.s. represents the expansion of the effective action in absence of external field (see Section 3), while the functional derivatives in the following terms represent the $\Lambda$-dependent susceptibility and the fermion-boson vertex in the different channels. Taking the derivative with respect to the scale parameter $\Lambda$ (see Appendix B) yields the following flow equations for the susceptibility and fermion-boson vertex (assuming SU(2) symmetry and momentum-frequency as well as spin conservation)

$$\partial_\Lambda \chi_{\mathrm{d/m}}^{nn',\Lambda}(q) = \int dk \left[ -S^\Lambda(k)\tilde{\gamma}_{4,\mathrm{d/m}}^{nn',\Lambda}(q,k) - \right.$$
$$\left. \gamma_{3,\mathrm{d/m}}^{n,\Lambda}(q,k)[G^\Lambda(k)S^\Lambda(q+k) + (S \leftrightarrow G)]\gamma_{3,\mathrm{d/m}}^{n',\Lambda,\dagger}(q,k) \right] \tag{14a}$$

$$\partial_\Lambda \chi_{\mathrm{sc}}^{nn',\Lambda}(q) = \int dk \left[ -S^\Lambda(k)\tilde{\gamma}_{4,\mathrm{sc}}^{nn',\Lambda}(q,k) + \right.$$
$$\left. \gamma_{3,\mathrm{sc}}^{n,\Lambda}(q,k)[G^\Lambda(k)S^\Lambda(q-k) + (S \leftrightarrow G)]\gamma_{3,\mathrm{sc}}^{n',\Lambda,\dagger}(q,k) \right], \tag{14b}$$

and respectively

$$\partial_\Lambda \gamma_{3,\mathrm{d/m}}^{n,\Lambda}(q,k) = \int dk' \left[ -S^\Lambda(k)\gamma_{5,\mathrm{d/m}}^{n,\Lambda}(q,k,k') \right.$$
$$\left. \gamma_{3,\mathrm{d/m}}^{n,\Lambda}(q,k')[G^\Lambda(k')S^\Lambda(q+k') + (S \leftrightarrow G)]\gamma_{4,\mathrm{d/m}}^\Lambda(q,k',k) \right] \tag{15a}$$

$$\partial_\Lambda \gamma_{3,\mathrm{sc}}^{n,\Lambda}(q,k) = \int dk' \left[ -S^\Lambda(k)\gamma_{5,\mathrm{sc}}^{n,\Lambda}(q,k,k') + \right.$$
$$\left. \gamma_{3,\mathrm{sc}}^{n,\Lambda}(q,k')[G^\Lambda(k')S^\Lambda(q-k') + (S \leftrightarrow G)]\gamma_{4,\mathrm{sc}}^\Lambda(q,k',k) \right], \tag{15b}$$

where

$$S^\Lambda = \partial_\Lambda G^\Lambda|_{\Sigma=\mathrm{const}} \tag{16}$$

represents the single-scale propagator. The function $\tilde{\gamma}_4$, differently from the (fermionic) two-particle vertex $\gamma_4$, represents a mixed bosonic-fermionic vertex, i.e., with two bosonic and two fermionic legs where we summed over its spin dependences

$$\tilde{\gamma}_{4,\eta}^{nn',\Lambda}(q,k) = \sum_\sigma \tilde{\gamma}_{4,\eta;\sigma\sigma}^{nn',\Lambda}(q,k), \tag{17}$$

while the spin-independent form for $\gamma_5$ used in Eqs. (15) reads

$$\gamma_{5,\mathrm{d/m}}^{n,\Lambda}(q,k,k') = \sum_{\sigma'} \sigma_{\sigma\sigma}^{0/3} \gamma_{5,\mathrm{d/m};\sigma\sigma\sigma'\sigma'}^{n,\Lambda}(q,k,k') \tag{18a}$$

$$\gamma_{5,\mathrm{sc}}^{n,\Lambda}(q,k,k') = \sum_{\sigma'} \gamma_{5,\mathrm{sc};\sigma\bar{\sigma}\sigma'\sigma'}^{n,\Lambda}(q,k,k'). \tag{18b}$$

The conventional approximations [1, 58, 60] disregard the first terms on the r.h.s. of Eqs. (14) and (15). This $1\ell$ approximation is consistent with the corresponding approximation of $\gamma_4^\Lambda$ (see Appendix C) and justified in the weak-coupling regime. Using the notation of Refs. [45, 55], one can rewrite the $1\ell$ approximation of Eqs. (14) and (15) in a more concise tensor-form

$$\dot{\boldsymbol{\chi}}_\eta^{\Lambda(1)} = \boldsymbol{\gamma}_{3,\eta}^\Lambda \circ \dot{\boldsymbol{\Pi}}_{S,\eta}^\Lambda \circ \boldsymbol{\gamma}_{3,\eta}^{\Lambda,\dagger} \tag{19a}$$

$$\dot{\boldsymbol{\gamma}}_{3,\eta}^{\Lambda(1)} = \boldsymbol{\gamma}_{3,\eta}^\Lambda \circ \dot{\boldsymbol{\Pi}}_{S,\eta}^\Lambda \circ \boldsymbol{\gamma}_{4,\eta}^\Lambda. \tag{19b}$$

where

$$\dot{\Pi}^\Lambda_{S,\mathrm{d/m}(ph)}(q,k) = -G^\Lambda(k)S^\Lambda(q+k) + (S \leftrightarrow G) \tag{20a}$$

$$\dot{\Pi}^\Lambda_{S,\mathrm{sc}(pp)}(q,k) = G^\Lambda(k)S^\Lambda(q-k) + (S \leftrightarrow G). \tag{20b}$$

We here introduced also the subscript $ph$ and $pp$ indicating the diagrammatic channels that will be referred to in Sec. 3.

So far we pinpointed two possible ways to compute the susceptibility and fermion-boson vertex from an fRG calculation: (i) Solving Eqs. (14) and (15) alongside the ones for $\Sigma$ and $\gamma_4$ (at the same level of approximation), and (ii) by means of Eqs. (8) and (10) at the end of the fRG flow, using $\Sigma^{\Lambda_{\mathrm{final}}}$ and $\gamma_4^{\Lambda_{\mathrm{final}}}$, later referred to as "post-processing". These two procedures are non-equivalent in the presence of approximations, e.g., if one restricts oneself to the $1\ell$ level. This leads to an ambiguity in practical implementations of the fRG. In fact, as shown in Appendix D, the two results deviate at $\mathcal{O}((\gamma_4^\Lambda)^2)$ for the $1\ell$ case (for a larger number of loops the deviations occur at higher orders in the effective interaction $\gamma_4^\Lambda$). In order to solve this ambiguity we note that the exact relations (8) and (10) are fulfilled in the PA. At the same time, the recently introduced multiloop extension allows one to sum up all parquet diagrams. Hence, generalizing the multiloop flow to the computation of the response functions recovers the equivalence of the two procedures.

In order to derive the mfRG equations for the response functions, we first recall the channel-decomposition of the two-particle vertex as known from the parquet formalism. The latter divides $\gamma_4$ in the two-particle reducible vertex $\phi$ (all diagrams that can be divided into two separate ones by removing two internal fermionic propagators) and the two-particle irreducible vertex $I$ (which can be not be divided). Depending on the direction of the propagation lines the diagrams are reducible in either parallel, longitudinal antiparallel or transverse antiparallel, corresponding to the particle-particle, particle-hole, and particle-hole crossed channel, respectively. Besides this diagrammatic channel decomposition, there is also a distinct physical channel decomposition that identifies the components $\eta = \{\mathrm{d}, \mathrm{m}, \mathrm{sc}\}$ and which we will use in the following. Inserting this decomposition into the flow equation for the two-particle vertex, we obtain

$$\partial_\Lambda \gamma^\Lambda_{4,\eta} = \partial_\Lambda I^\Lambda_\eta + \partial_\Lambda \phi^\Lambda_\eta. \tag{21}$$

While the usual diagrammatic channel decomposition [64] leads to simple expressions for the two-particle irreducible vertex $I^\Lambda_\eta$, the latter assumes a more complicated form in the physical channels

$$I^\Lambda_\mathrm{d}(q,k,k') = -U - \frac{1}{2}\phi^\Lambda_\mathrm{d}(k'-k,k,k+q) - \frac{3}{2}\phi^\Lambda_\mathrm{m}(k'-k,k,k+q) + $$
$$+ 2\phi^\Lambda_\mathrm{sc}(q+k+k',k,k') - \phi^\Lambda_\mathrm{sc}(q+k+k',k,q+k) \tag{22a}$$

$$I^\Lambda_\mathrm{m}(q,k,k') = U - \frac{1}{2}\phi^\Lambda_\mathrm{d}(k'-k,k,k+q) + \frac{1}{2}\phi^\Lambda_\mathrm{m}(k'-k,k,k+q) - $$
$$+ \phi^\Lambda_\mathrm{sc}(q+k+k',k,k+q) \tag{22b}$$

$$I^\Lambda_\mathrm{sc}(q,k,k') = -U - \phi^\Lambda_\mathrm{m}(k'-k,k,q-k') + \frac{1}{2}\phi^\Lambda_\mathrm{d}(q-k-k',k,k') - $$
$$- \frac{1}{2}\phi^\Lambda_\mathrm{m}(q-k-k',k,k'), \tag{22c}$$

where we approximated the fully two-particle irreducible vertex by its first-order contribution in the interaction $\sim U$, which is known as PA.

We now derive the mfRG flow equations for the response functions, which mimic the effect of the mixed fermion-boson vertices $\tilde{\gamma}^\Lambda_4$ and $\gamma^\Lambda_5$ in the exact flow Eqs. (14) and (15). First, one performs the so-called Katanin substitution [45] $S^\Lambda \to \partial_\Lambda G^\Lambda$, which implies $\dot{\Pi}^\Lambda_{S,\eta} \to \dot{\Pi}^\Lambda_\eta$ in

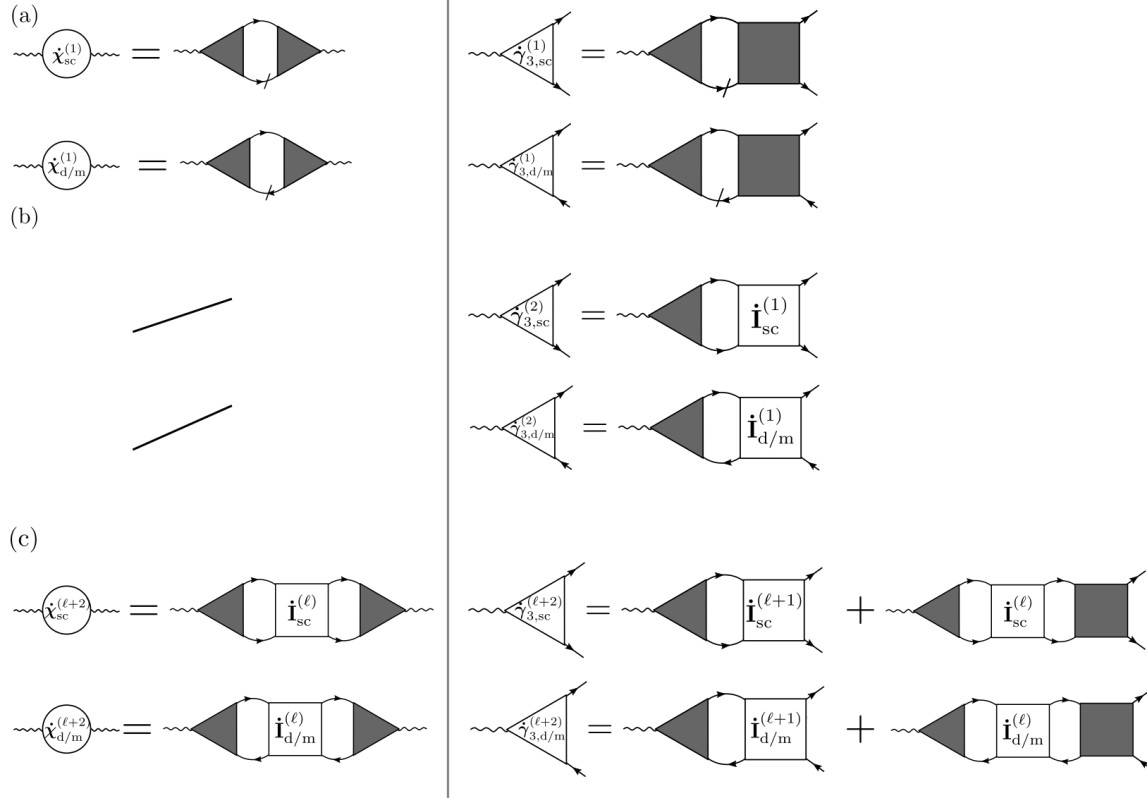

Figure 1: Multiloop flow equations for the susceptibility (left column) and the fermion-boson vertex (right column) for all physical channels $\eta = \{sc, m, d\}$. Whereas the filled boxes and triangles represent the vertex $\gamma^\Lambda_{4,\eta}$ and $\gamma^\Lambda_{3,\eta}$, respectively, the empty ones contain the scale-parameter derivative of the two-particle irreducible vertex $\dot{\mathbf{I}}^\Lambda_\eta$ in the respective channel (see Eq. (22)). (a) Standard one-loop truncated flow equations as in Eq. (19). (b) Two loop corrections for the fermion-boson vertex as in Eq. (24). As argued in the text, because of the fermionic leg contractions, no two-loop correction terms appear in the susceptibility flow equation. (c) Higher loop corrections starting from the third loop order for both susceptibility and fermion-boson vertex as reported in Eqs. (26) and (25), respectively.

the 1$\ell$ flow equations (19). One observes that all differentiated lines in these flow equations come from $\dot{\mathbf{\Pi}}^\Lambda_\eta$. Secondly, differentiated lines from the other channels are contained in the higher-loop terms of the expansion

$$\partial_\Lambda \boldsymbol{\chi}^\Lambda_\eta = \sum_{\ell \geqslant 1} \dot{\boldsymbol{\chi}}^{\Lambda(\ell)}_\eta \tag{23a}$$

$$\partial_\Lambda \boldsymbol{\gamma}^\Lambda_{3,\eta} = \sum_{\ell \geqslant 1} \dot{\boldsymbol{\gamma}}^{\Lambda(\ell)}_{3,\eta} . \tag{23b}$$

Using the channel decomposition (21), we can directly write down the 2$\ell$ correction to the flow of the fermion-boson vertex, which accounts for the leading-order diagrams of the effective interaction and stem from $\gamma^\Lambda_5$ in Eq. (15) (see Appendix E)

$$\dot{\boldsymbol{\gamma}}^{\Lambda(2)}_{3,\eta} = \boldsymbol{\gamma}^\Lambda_{3,\eta} \circ \mathbf{\Pi}^\Lambda_\eta \circ \dot{\mathbf{I}}^{\Lambda(1)}_\eta . \tag{24}$$

On the three- and higher-loop level, we can now use $\dot{\mathbf{I}}^{\Lambda(\ell)}_\eta$ in an analogous way. In addition,

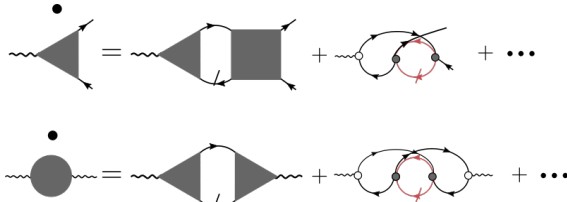

Figure 2: Multiloop corrections (beyond $1\ell$) for $\gamma_{3,\text{d/m}}^{\Lambda}$ (top) and $\chi_{\text{d/m}}^{\Lambda}$ (bottom) at the leading order in the bare interaction (filled black dot). The empty dot represents the bare fermion-boson vertex $\gamma_{3,\eta,0}^n(q,k) = f_n(\mathbf{k})$.

we have to consider the vertex corrections to the right of the differentiated lines, yielding

$$\dot{\gamma}_{3,\eta}^{\Lambda(\ell+2)} = \gamma_{3,\eta}^{\Lambda} \circ \Pi_{\eta}^{\Lambda} \circ \dot{\mathbf{I}}_{\eta}^{\Lambda(\ell+1)} + \gamma_{3,\eta}^{\Lambda} \circ \Pi_{\eta}^{\Lambda} \circ \dot{\mathbf{I}}_{\eta}^{\Lambda(\ell)} \circ \Pi_{\eta}^{\Lambda} \circ \gamma_{4,\eta}^{\Lambda} \,. \tag{25}$$

Considering the $1\ell$ flow equation of the susceptibility (19a), we see that the fermion-boson vertices provide vertex corrections on both sides of the differentiated lines in $\dot{\Pi}_{\eta}^{\Lambda}$. Hence, for all higher-loop corrections we can simply connect $\dot{\mathbf{I}}_{\eta}^{\Lambda(\ell)}$ to both fermion-boson vertices, thereby raising the loop order by two. We obtain $\dot{\chi}_{\eta}^{\Lambda(2)} = 0$, as well as

$$\dot{\chi}_{\eta}^{\Lambda(\ell+2)} = \gamma_{3,\eta}^{\Lambda} \circ \Pi_{\eta}^{\Lambda} \circ \dot{\mathbf{I}}_{\eta}^{\Lambda(\ell)} \circ \Pi_{\eta}^{\Lambda} \circ \gamma_{3,\eta}^{\Lambda,\dagger} \,. \tag{26}$$

For a schematic representation of the multiloop flow equations for $\chi_{\eta}$ and $\gamma_{3,\eta}$ see Fig. 1, while an example of the multiloop corrections at the leading order in the bare interaction is illustrated in Fig. 2. The above equations, together with the multiloop flow of the fermionic two-particle vertex (see Section 3.2) allow us to sum up all differentiated parquet diagrams of $\gamma_3^{\Lambda}$ and $\chi^{\Lambda}$. As a consequence, the aforementioned two ways of computing the response functions within the fRG become equivalent. We finally note that for a consistent fRG scheme, it is important to adopt the same level of approximation (truncating the sums in Eq. (23a) to a certain finite $\ell$-loop level) for all flowing quantities.

# 3 Numerical implementation

## 3.1 Full frequency and momentum parametrization

In order to illustrate the fRG algorithm adopted in the present work, let us start from the flow equations for the 1PI fermionic vertex in the $1\ell$ fRG approximation. In the following, the SU(2) spin conserving symmetry will be always assumed. Exploiting this symmetry, the self-energy and two-particle fermionic vertices can be written as

$$\Sigma_{\sigma\sigma'}(k) = \delta_{\sigma,\sigma'}\Sigma_{\sigma}(k) = \delta_{\sigma,\sigma'}\Sigma(k) \tag{27}$$

$$\gamma_{4,\sigma_1\sigma_2\sigma_3\sigma_4}(k_1,k_2,k_3) = [-\delta_{\sigma_1,\sigma_4}\delta_{\sigma_2,\sigma_3}\gamma_4(k_1,k_4,k_3) + \delta_{\sigma_1,\sigma_2}\delta_{\sigma_3,\sigma_4}\gamma_4(k_1,k_2,k_3)]\,, \tag{28}$$

where the fourth argument of $\gamma_4$ is determined by $k_4 = k_1 + k_3 - k_2$ in a momentum and energy conserving system. The spin-independent flow equation for the self-energy reads

$$\dot{\Sigma}^{\Lambda}(k) = -\int dp\, S^{\Lambda}(p)\left[2\,\gamma_4^{\Lambda}(k,k,p) - \gamma_4^{\Lambda}(p,k,k)\right], \tag{29}$$

where $S^{\Lambda}(p)$ represents the single-scale propagator specified in Eq. (16). We formulate the flow equation for $\gamma_4$ in the channel decomposed form suggested by Husemann and Salmhofer

[28]

$$\dot\gamma_4^\Lambda(k_1,k_2,k_3) = \mathcal{T}_{pp}^\Lambda(k_1+k_3,k_1,k_4) + \mathcal{T}_{ph}^\Lambda(k_2-k_1,k_1,k_4) + \mathcal{T}_{\overline{ph}}^\Lambda(k_3-k_2,k_1,k_2)\,, \tag{30}$$

where the diagrammatic channel index $r = \{pp, ph, \overline{ph}\}$ distinguishes between particle-particle, particle-hole and particle-hole exchange diagrams, and the first dependence of the functions $\mathcal{T}_r^\Lambda$ refers to the bosonic four-momentum transfer in the internal loop of their corresponding equations

$$\mathcal{T}_{pp}^\Lambda(k_1+k_3,k_1,k_4) = \int dp\,\gamma_4^\Lambda(k_1,k_1+k_3-p,k_3)\gamma_4^\Lambda(p,k_2,k_1+k_3-p)\times$$
$$\left[S^\Lambda(p)G^\Lambda(k_1+k_3-p) + (S\leftrightarrow G)\right], \tag{31a}$$

$$\mathcal{T}_{ph}^\Lambda(k_2-k_1,k_1,k_4) = -\int dp\Big[2\gamma_4^\Lambda(k_1,k_2,k_2-k_1+p)\,\gamma_4^\Lambda(p,k_2-k_1+p,k_3)-$$
$$\gamma_4^\Lambda(k_1,p,k_2-k_1+p)\,\gamma_4^\Lambda(p,k_2-k_1+p,k_3)-$$
$$\gamma_4^\Lambda(k_1,k_2,k_2-k_1+p)\,\gamma_4^\Lambda(p,k_2,k_3)\Big]\times$$
$$\left[S^\Lambda(p)G^\Lambda(k_2-k_1+p) + (S\leftrightarrow G)\right], \tag{31b}$$

$$\mathcal{T}_{\overline{ph}}^\Lambda(k_3-k_2,k_1,k_2) = \int dp\,\gamma_4^\Lambda(k_1,p,k_3-k_2+p)\gamma_4^\Lambda(p,k_2,k_3)\times$$
$$\left[S^\Lambda(p)G^\Lambda(k_3-k_2+p) + (S\leftrightarrow G)\right]. \tag{31c}$$

Note that the assignment of the various terms on the right hand side of the flow equation to the three channels is not unique. The version we use here corresponds to the choice by Wang et al. in their singular-mode fRG [31, 65]. Each of the above equations depends, besides the aforementioned bosonic transfer dependence ($k_1+k_3$, $k_2-k_1$ and $k_3-k_2$), on two fermionic dependencies. Such mixed 'bosonic-fermionic' notation, referred to as 'non-symmetrized' notation, has been substituted in some work (e.g., in Ref. [31]) by a different notation where the dependencies of the four fermionic propagators involved in the scattering process have been chosen symmetrically with respect to the bosonic four-momentum transfer. This symmetrized notation simplifies the implementation of the symmetries exploited in the fRG code (see Appendix F and Ref. [31]) but leads to less compact flow equations. The equation (31) generates the two-particle reducible vertices $\mathcal{T}_r = \dot\phi_r$ of the diagrammatic parquet decomposition

$$\gamma_4(k_1,k_2,k_3) \approx U + \phi_{pp}(k_1+k_3,k_1,k_4) + \phi_{ph}(k_2-k_1,k_1,k_4) + \phi_{\overline{ph}}(k_3-k_2,k_1,k_2)\,. \tag{32}$$

The two-particle fermionic vertex can be reconstructed by using Eq. (32). The use of a mixed 'bosonic-fermionic' notation allows us to identify the bosonic transfer four-momentum as the strongest dependence, while the two fermionic dependencies can be treated with controllable approximations. In the following we illustrated two efficient ways to simplify the treatment of both momentum and frequency dependencies.

### 3.1.1 Truncated Unity fRG

The approximation for the fermionic momentum dependencies in TUfRG [31] is done by the expansion of the fermionic momentum dependencies in form factors, illustrated here for the $pp$ channel

$$\phi_{pp}(\mathbf{q},\mathbf{k},\mathbf{k}') = \sum_{n,n'} f_n(\mathbf{k})f_{n'}^*(\mathbf{k}')P_{n,n'}(\mathbf{q})\,, \tag{33}$$

while the expansion of the $\phi_{ph}$ and $\phi_{\overline{ph}}$ analogously defines $D_{n,n'}(\mathbf{q})$ and $C_{n,n'}(\mathbf{q})$. Following the conventions introduced in previous works [28, 29, 31, 41, 66, 67], we choose the form factors such that they correspond to a specific shell of neighbors in the real space lattice. The unity inserted in the flow equations contains a complete basis set of form factors

$$\mathbb{1} = \int d\mathbf{p}' \sum_n f_n^*(\mathbf{p}')f_n(\mathbf{p}) \,. \tag{34}$$

Converged results can be obtained already with a small set of form factors [31], i.e. the unity (34) can be approximated by a truncated unity, giving rise to the name "truncated-unity fRG". For a fast convergence it is convenient to include one shell after another, starting from the constant local form factor and increasing the distance of neighbors taken into account. The form factors used in this paper are listed in Table 1.

A major difficulty in this approach is the feedback of the different channels into each other. In addition to the dressing of the objects by the form factors, the translation of the notation in momentum and frequency from one to another channel has to be considered. Computationally time consuming integrations in momentum space can be avoided by Fourier transformation and evaluation in real space [31, 65]. Furthermore the expression of the projection in terms of a matrix multiplication allows for the precalculation of the projection matrices which can be found in the Appendix F.

### 3.1.2 Dynamical fRG

In frequency space, we adopt the simplifications proposed in Refs. [33, 34]. For all systems with an instantaneous microscopic interaction one can use diagrammatic arguments to prove that, in the high-frequency regime, the fermionic two-particle vertex exhibits a simplified asymptotic structure. In this region one can reduce the three-dimensional frequency dependence of $\gamma_4$ using functions with a simplified parametric dependence. It is straightforward to see that, sending all three frequencies to infinity, $\gamma_4$ reduces to the instantaneous microscopic interaction, which in the present case is represented by the Hubbard on-site $U$. The contribution of the reducible vertices $\phi_r$ to $\gamma_4$ becomes non-negligible if the bosonic frequency transfer is kept finite, while sending the two secondary fermionic frequencies to infinity. This contribution, depending on a single bosonic frequency transfer in a given channel $r$, is denoted by $\mathcal{K}_{1,r}(i\omega_l, q)$. For models with an instantaneous and local microscopic interaction, one observes that the momentum dependencies disappear alongside the frequency dependencies when performing such limits. In the limit where just one fermionic frequency is sent to infinity, the vertex $\phi_r$ can be parametrized by the function $\mathcal{K}_{2,r}(i\omega_l, i\nu_o, q, k) + \mathcal{K}_{1,r}(i\omega_l, q)$. By subtracting the asymptotic functions from the full object $\phi_r$ we obtain the so-called [34]"rest-function" $\mathcal{R}(i\omega_l, i\nu_o, i\nu_{o'}, q, k, k')$ which decays to zero within a small frequency box. The parametrization in terms of $\mathcal{K}_{1/2}$ allows us to reduce the numerical cost of computing and storing the fermionic two-particle vertices. In fact, for any of the three channels, we calculate the fRG flow of the three-frequency dependent function $\mathcal{R}$ on a small low-frequency region and add the information on the high frequencies by computing the flow of the functions $\mathcal{K}_1$ and $\mathcal{K}_2$ which are numerically less demanding. The full two-particle reducible vertex $\phi_r$ is then recovered by

$$\begin{aligned}
\phi_r(i\omega_l, i\nu_o, i\nu_{o'}, \mathbf{q}, \mathbf{k}, \mathbf{k}') = &\mathcal{R}_r(i\omega_l, i\nu_o, i\nu_{o'}, \mathbf{q}, \mathbf{k}, \mathbf{k}') + \\
&\mathcal{K}_{2,r}(i\omega_l, i\nu_o, \mathbf{q}, \mathbf{k}) + \bar{\mathcal{K}}_{2,r}(i\omega, i\nu_{o'}, \mathbf{q}, \mathbf{k}') + \mathcal{K}_{1,r}(i\omega_l, \mathbf{q}),
\end{aligned} \tag{35}$$

where $\bar{\mathcal{K}}_{2,r}$ can be obtained from $\mathcal{K}_{2,r}$ by exploiting the time reversal symmetry (see Appendix A.3).

### 3.1.3 Flow equations for the TU-dynamical fRG

Finally, applying the aforementioned projection on the form-factor basis we can write matrix-like $1\ell$ fRG flow equations for the self-energy, the two-particle vertex, the fermion-boson vertex and the susceptibility:

$$\dot{\Sigma}^{\Lambda}(k) = -\int dp\, S^{\Lambda}(p) \Big[ 2\, \gamma_4^{\Lambda}(k,k,p) - \gamma_4^{\Lambda}(p,k,k) \Big] \tag{36a}$$

$$\dot{\mathbf{P}}^{\Lambda}(q,i\nu_o,i\nu_{o'}) = \frac{1}{\beta} \sum_{i\nu_{n''}} \boldsymbol{\gamma}_{4,P}^{\Lambda}(q,i\nu_o,i\nu_{n''}) \dot{\boldsymbol{\Pi}}_{S,pp}^{\Lambda}(q,i\nu_{n''}) \boldsymbol{\gamma}_{4,P}^{\Lambda}(q,i\nu_{n''},i\nu_{o'}) \tag{36b}$$

$$\dot{\mathbf{D}}^{\Lambda}(q,i\nu_o,i\nu_{o'}) = \frac{1}{\beta} \sum_{i\nu_{n''}} \dot{\boldsymbol{\Pi}}_{S,ph}^{\Lambda}(q,i\nu_{n''}) \Big[ 2\boldsymbol{\gamma}_{4,D}^{\Lambda}(q,i\nu_o,i\nu_{n''}) \boldsymbol{\gamma}_{4,D}^{\Lambda}(q,i\nu_{n''},i\nu_{o'}) -$$
$$-\boldsymbol{\gamma}_{4,C}^{\Lambda}(q,i\nu_o,i\nu_{n''}) \boldsymbol{\gamma}_{4,D}^{\Lambda}(q,i\nu_{n''},i\nu_{o'}) -$$
$$-\boldsymbol{\gamma}_{4,D}^{\Lambda}(q,i\nu_o,i\nu_{n''}) \boldsymbol{\gamma}_{4,C}^{\Lambda}(q,i\nu_{n''},i\nu_{o'}) \Big] \tag{36c}$$

$$\dot{\mathbf{C}}^{\Lambda}(q,i\nu_o,i\nu_{o'}) = -\frac{1}{\beta} \sum_{i\nu_{n''}} \boldsymbol{\gamma}_{4,C}^{\Lambda}(q,i\nu_o,i\nu_{n''}) \dot{\boldsymbol{\Pi}}_{S,ph}^{\Lambda}(q,i\nu_{n''}) \boldsymbol{\gamma}_{4,C}^{\Lambda}(q,i\nu_{n''},i\nu_{o'}) \tag{36d}$$

$$\dot{\boldsymbol{\gamma}}_{3,\eta}^{\Lambda}(q,i\nu_o) = \frac{1}{\beta} \sum_{i\nu_{n'}} \boldsymbol{\gamma}_{3,\eta}^{\Lambda}(q,i\nu_{n'}) \dot{\boldsymbol{\Pi}}_{S,\eta}^{\Lambda}(q,i\nu_{n'}) \boldsymbol{\gamma}_{4,\eta}^{\Lambda}(q,i\nu_{n'},i\nu_o) \tag{36e}$$

$$\dot{\boldsymbol{\chi}}_{\eta}^{\Lambda}(q) = \frac{1}{\beta} \sum_{i\nu_n} \boldsymbol{\gamma}_{3,\eta}^{\Lambda}(q,i\nu_n) \dot{\boldsymbol{\Pi}}_{S,\eta}^{\Lambda}(q,i\nu_n) \boldsymbol{\gamma}_{3,\eta}^{\Lambda}(q,i\nu_n) \,, \tag{36f}$$

where the multiplication of bold symbols has here to be understood as matrix multiplications with respect to the form factors. For a schematic visualization of the practical implementation of these equations, see Fig. 3. We note that, in order to derive Eqs. (36), we inserted the unity (34), truncated to a finite number of form factors, in Eqs. (19) as well as in (31). The full vertex $\gamma_{4,r}$, with $r = \{P, D, C\}$ represents the fermionic two-particle vertex in the channel-specific mixed 'bosonic-fermionic' notations, while $\gamma_{4,\eta}$ with $\eta = \{\text{sc}, \text{d}, \text{m}\}$ is given by

$$\gamma_{4,\text{d}} = 2\gamma_{4,D} - \gamma_{4,C} \tag{37a}$$
$$\gamma_{4,\text{m}} = -\gamma_{4,D} \tag{37b}$$
$$\gamma_{4,\text{sc}} = \gamma_{4,P} \,. \tag{37c}$$

Note that the TUfRG equations for the channel couplings $\mathbf{P}^{\Lambda}(q,i\nu_o,i\nu_{o'})$, $\mathbf{D}^{\Lambda}(q,i\nu_o,i\nu_{o'})$, and $\mathbf{C}^{\Lambda}(q,i\nu_o,i\nu_{o'})$ are equivalent to the singular-mode fRG equations derived earlier in a different way by Wang et al. [65], as also discussed in [31]. The new point here is the dynamical implementation also taking into account the frequency dependence.

The $1\ell$-fRG flow consists in integrating the coupled differential equations in (36) with the following initial conditions:

$$\Sigma^{\Lambda_{\text{init}}} = 0 \tag{38a}$$
$$\boldsymbol{\gamma}_{4,P}^{\Lambda_{\text{init}}} = \boldsymbol{\gamma}_{4,D}^{\Lambda_{\text{init}}} = \boldsymbol{\gamma}_{4,C}^{\Lambda_{\text{init}}} = U\delta_{n,0}\delta_{n',0} \tag{38b}$$
$$\boldsymbol{\chi}_{\eta}^{\Lambda_{\text{init}}} = 0 \tag{38c}$$
$$\boldsymbol{\gamma}_{3,\eta}^{\Lambda_{\text{init}}} = \delta_{n,n'} \,. \tag{38d}$$

Finally, $\dot{\boldsymbol{\Pi}}_{S,\eta}$ relative to the particle-hole $\eta = \{\text{d}/\text{m}(ph)\}$ and to the particle-particle channels

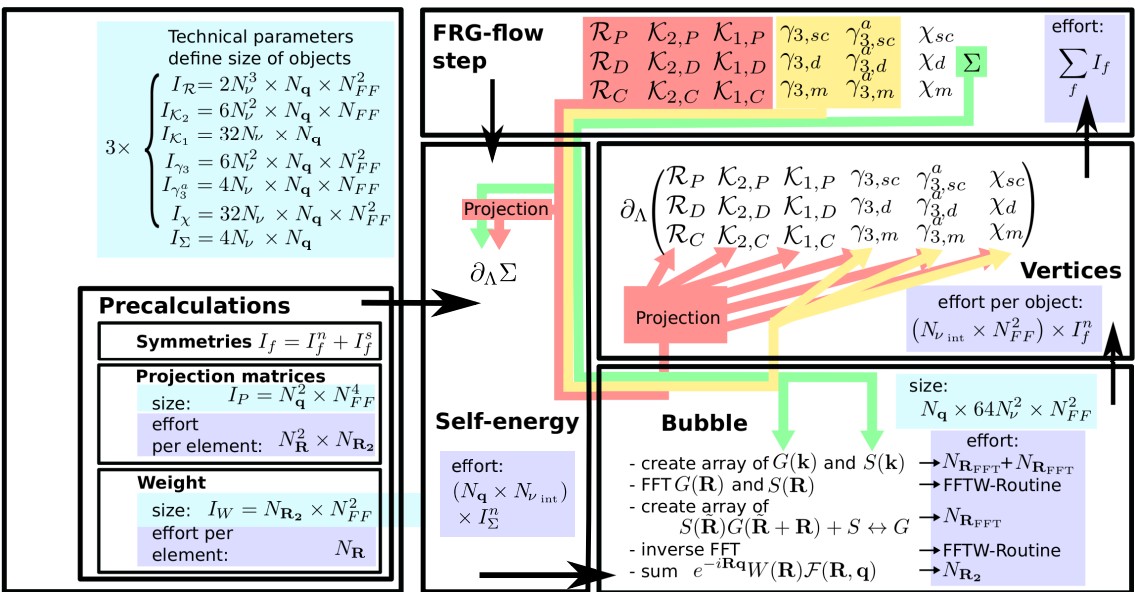

Figure 3: Schematic code structure specifying the array sizes and the numerical effort of the single steps. $I_f$ denotes the number of elements of the object $f$. $N_\nu$ is the number of fermionic frequencies of the rest function, $N_{\mathbf{q}}$ the number of bosonic momentum patches, $N_{FF}$ the number of form factors, $N_{\mathbf{R}_{FFT}}$ and $N_{\nu_{int}}$ the number of frequencies over which the internal fermionic bubble is integrated. The symmetries reduce the total number of elements $I_f$ to $I_f^n$ independent elements which have to be calculated and to $I_f^s$ which can be obtained by using symmetry relations. The arrows indicate the feedback of the different parts, namely the two-particle fermionic vertices (red), fermion-boson vertex (yellow), and the self-energy (green). In the multiloop-extended version of the fRG program, the numerical effort scales linearly in the number of loops $\ell$ accounted. Here, $\gamma_{3,\eta}^a$ is the asymptotic function of $\gamma_{3,\eta}$, obtained by sending the fermionic frequency to infinity.

$\eta = \{\text{sc}(pp)\}$, are defined as

$$\dot{\Pi}_{S,\text{d/m}(ph)}^\Lambda(i\omega_l, i\nu_o, \mathbf{q})_{n,n'} = -\int d\mathbf{p} f_n^*(\mathbf{p}) f_{n'}(\mathbf{p})\, \dot{\Pi}_{S,\text{d/m}(ph)}^\Lambda(i\omega_l, i\nu_o, \mathbf{q}, \mathbf{p})\,, \tag{39a}$$

$$\dot{\Pi}_{S,\text{sc}(pp)}^\Lambda(i\omega_l, i\nu_o, \mathbf{q})_{n,n'} = \int d\mathbf{p} f_n^*(\mathbf{p}) f_{n'}(\mathbf{p})\, \dot{\Pi}_{S,\text{sc}(pp)}^\Lambda(i\omega_l, i\nu_o, \mathbf{q}, \mathbf{p})\,, \tag{39b}$$

where $\dot{\Pi}_{S,\eta}^\Lambda(q,k)$ is defined in Eq. (20). In order to perform the momentum integration in Eqs. (39) we adopt a strategy which, exploiting the convolution theorem, represents a numerically convenient alternative to the use of adaptive integration algorithms. The latter is described in the following section.

### 3.1.4 Calculation of the fermionic particle-hole and particle-particle excitation

We here present a numerically convenient way of calculating the fermionic particle-hole and particle-particle bubbles in the flow equations of the vertex (36), defined in Eqs. (39). Since the integral over momenta is very sensitive on the momentum mesh resolution near the Fermi surface and a refined adaptive integration is computationally time consuming, we rewrote the integrals in such a way to use the convolution theorem. The Green's function can then be transformed via the Fast-Fourier-Transform (FFT) to real space, where the real-space expression of the form factors is provided in Table 1. After some algebraic steps, we find an

expression without momentum integration

$$\dot{\Pi}_{S,ph}(i\omega_l, i\nu_o, \mathbf{q})]_{n,n'} = -\sum_{\mathbf{R}} e^{i\mathbf{R}\mathbf{q}} W_{n,n'}(\mathbf{R}) \times$$

$$\mathcal{F}\Big[S(i\nu_o, -\tilde{\mathbf{R}})G(i\omega_l + i\nu_o, \tilde{\mathbf{R}} - \mathbf{R}) + (S \leftrightarrow G)\Big](\mathbf{q}), \tag{40a}$$

$$\dot{\Pi}_{S,pp}(i\omega_l, i\nu_o, \mathbf{q})]_{n,n'} = \sum_{\mathbf{R}} e^{-i\mathbf{R}\mathbf{q}} W_{n,n'}(\mathbf{R}) \times$$

$$\mathcal{F}\Big[S(i\nu_o, \tilde{\mathbf{R}})G(i\omega_l - i\nu_o, \tilde{\mathbf{R}} + \mathbf{R}) + (S \leftrightarrow G)\Big](\mathbf{q}), \tag{40b}$$

where $\mathcal{F}\Big[f(\tilde{R})\Big](\mathbf{k})$ is the Fourier transform which can be determined by using FFT-methods and the weight $W_{n,n'}(R)$ is defined as

$$W_{n,n'}(\mathbf{R}) = \sum_{\mathbf{R}'} f_n^*(\mathbf{R}') f_{n'}(\mathbf{R} + \mathbf{R}'). \tag{41}$$

The infinite sum of the lattice points in Eqs. (40b), (40a), and (41) is restricted by the finite range of the form factors for a specific truncation. For instance the sum in Eq. (41) is limited to the maximal shell taken into account by the form factors. Hence, the weight has a nonzero contribution only inside a shell twice as large the maximal shell of the form factors and therefore the sum in Eq. (40a) can be constrained to twice the distance of the maximal form factor shell.

The momentum and real space grid for the Fourier transformations needed in the bubbles has to be chosen fine enough, especially at low temperatures. The convergence in terms of FFT-grid points $N_{\mathbf{R}_{FFT}}$ has to be checked separately from the bosonic momentum grid of the vertex. Recent works using the TUfRG [31, 67] have demonstrated that, if needed, both low temperatures and high wavevector resolutions can be achieved by means of an adaptive integration scheme.

### 3.1.5 Diagrammatic and lattice related symmetries

Further numerical simplifications come from the extensive use of symmetries related to diagrammatic arguments and lattice-specific properties, which can be found in Appendix A.

## 3.2 The mfRG implementation

The mfRG flow introduced in Ref. [55] ameliorates the approximation induced by the truncation of the fRG hierarchy of flow equations as it incorporates all contributions from the six-point vertex $\gamma_6$ that can be computed at the same cost as the $1\ell$ flow considered so far. In fact, it includes all contributions coming from $\gamma_6$ that can be computed in an iterative $1\ell$ construction of four-point objects; hence, the numerical effort grows only linearly in the number of loops retained. It has been shown [55] that the multiloop prescription fully sums up all parquet diagrams. This gives rise to a number of advantageous properties, the most important of which are (i) that the multiloop corrections restore the independence on the choice of regulator, and (ii) that the multiloop flow fully accounts for the interplay between different two-particle channels and thus hampers spurious vertex divergences coming from ladder diagrams in the individual channels.

Let us briefly recall the multiloop vertex flow employing the same line of arguments as used for the flow of the response functions in Section 2.2. We consider the reducible vertices in the physical channels $\phi_{\eta=\{sc,d,m\}}$. At first, one performs the Katanin substitution [45] $S^\Lambda \to \partial_\Lambda G^\Lambda$ ($\dot{\Pi}_{S,\eta} \to \dot{\Pi}_\eta$) in the $1\ell$ flow equation

$$\dot{\phi}_\eta^{\Lambda,(1)} = \gamma_{4,\eta}^\Lambda \circ \dot{\Pi}_\eta^\Lambda \circ \gamma_{4,\eta}^\Lambda, \tag{42}$$

and finds that, for every channel $\boldsymbol{\phi}_\eta^\Lambda$, all differentiated lines come from $\dot{\boldsymbol{\Pi}}_\eta^\Lambda$. Differentiated lines from the other channels are contained in higher-order terms of the loop expansion

$$\partial_\Lambda \boldsymbol{\phi}_\eta^\Lambda = \sum_{\ell \geq 1} \dot{\boldsymbol{\phi}}_\eta^{\Lambda,(\ell)} \ . \tag{43}$$

Using the channel decomposition (21), one has the two-loop correction

$$\dot{\boldsymbol{\phi}}_\eta^{\Lambda,(2)} = \boldsymbol{\gamma}_{4,\eta}^\Lambda \circ \boldsymbol{\Pi}_\eta^\Lambda \circ \dot{\mathbf{i}}_\eta^{\Lambda,(1)} + \dot{\mathbf{i}}_\eta^{\Lambda,(1)} \circ \boldsymbol{\Pi}_\eta^\Lambda \circ \boldsymbol{\gamma}_{4,\eta}^\Lambda \ , \tag{44}$$

where, according to Eq. (22), $\dot{\mathbf{i}}_\eta^{\Lambda,(\ell)}$ can be determined from the $\dot{\boldsymbol{\phi}}_{\eta'}^{\Lambda,(\ell)}$ of the complementary channels $\eta' \neq \eta$. All higher-loop terms are obtained in a similar fashion where one additionally accounts for vertex corrections to both sides of $\dot{\mathbf{i}}_\eta^{\Lambda,(\ell)}$

$$\begin{aligned}
\dot{\boldsymbol{\phi}}_\eta^{\Lambda,(\ell+2)} &= \boldsymbol{\gamma}_{4,\eta}^\Lambda \circ \boldsymbol{\Pi}_\eta^\Lambda \circ \dot{\mathbf{i}}_\eta^{\Lambda,(\ell+1)} + \dot{\mathbf{i}}_\eta^{\Lambda,(\ell+1)} \circ \boldsymbol{\Pi}_\eta^\Lambda \circ \boldsymbol{\gamma}_{4,\eta}^\Lambda + \boldsymbol{\gamma}_{4,\eta}^\Lambda \circ \boldsymbol{\Pi}_\eta^\Lambda \circ \dot{\mathbf{i}}_\eta^{\Lambda,(\ell)} \circ \boldsymbol{\Pi}_\eta^\Lambda \circ \boldsymbol{\gamma}_{4,\eta}^\Lambda \\
&= \left(\dot{\boldsymbol{\phi}}_\eta^{\Lambda,(\ell+2)}\right)_{\mathrm{R}} + \left(\dot{\boldsymbol{\phi}}_\eta^{\Lambda,(\ell+2)}\right)_{\mathrm{L}} + \left(\dot{\boldsymbol{\phi}}_\eta^{\Lambda,(\ell+2)}\right)_{\mathrm{C}} \ ,
\end{aligned} \tag{45}$$

where in the last line the subscripts $\{\mathrm{R, L, C}\}$ refer to the diagrammatic position of $\dot{\mathbf{i}}^\Lambda$, i.e., right, left and central, respectively. Using Eq. (21) one can easily deduce the multiloop flow of the vertices $\boldsymbol{\gamma}_{4,\eta}$

$$\partial_\Lambda \boldsymbol{\gamma}_{4,\eta}^\Lambda = \sum_{\ell \geq 1} \dot{\boldsymbol{\gamma}}_{4,\eta}^{\Lambda,(\ell)} = \sum_{\ell \geq 1} \left(\dot{\boldsymbol{\phi}}_\eta^{\Lambda,(\ell)} + \dot{\mathbf{i}}_\eta^{\Lambda,(\ell)}\right) \ . \tag{46}$$

In Ref. [55], it has further been pointed out that corrections to the self-energy flow (29) are necessary in order to generate all differentiated diagrams of the parquet self-energy. These corrections are included in the central part of the vertex flow $\boldsymbol{\gamma}_{4,\eta} \circ \boldsymbol{\Pi}_\eta \circ \dot{\mathbf{i}}_\eta \circ \boldsymbol{\Pi}_\eta \circ \boldsymbol{\gamma}_{4,\eta}$ and read

$$\partial_\Lambda \Sigma^\Lambda = \dot{\Sigma}^\Lambda + \delta \dot{\Sigma}_1^\Lambda + \delta \dot{\Sigma}_2^\Lambda \ , \tag{47}$$

with $\dot{\Sigma}$ given by Eq. (36a) and

$$\delta \dot{\Sigma}_1^\Lambda(k) = -\int dp \, G^\Lambda(p) \left[ 2\left(\dot{\boldsymbol{\phi}}_{\bar{D}}^\Lambda\right)_{\mathrm{C}}(k,p,k) - \left(\dot{\boldsymbol{\phi}}_{\bar{D}}^\Lambda\right)_{\mathrm{C}}(p,k,k) \right] \tag{48a}$$

$$\delta \dot{\Sigma}_2^\Lambda(k) = -\int dp \, \delta S^\Lambda(p) \left[ 2\boldsymbol{\gamma}_4^\Lambda(k,p,k) - \boldsymbol{\gamma}_4^\Lambda(p,k,k) \right] \ , \tag{48b}$$

where the central part (see Eq. (45)) for the differentiated reducible vertices $\dot{\boldsymbol{\phi}}_{r=\{P,C,D\}} = \{\dot{P}, \dot{D}, \dot{C}\}$ is defined by

$$\begin{aligned}
\left(\dot{\boldsymbol{\phi}}_{\bar{D}}^\Lambda\right)_{\mathrm{C}}(k_1,k_2,k_3) = \sum_{\ell \geq 1} \sum_{n,n'} \Big[ & f_n(\mathbf{k}_1) f_{n'}^*(\mathbf{k}_4) \left(\dot{\boldsymbol{\phi}}_P^{\Lambda,(\ell)}\right)_{\mathrm{C}}^{n,n'}(\nu_1 + \nu_3, \nu_1, \nu_4, \mathbf{k}_1 + \mathbf{k}_3) + \\
& f_n(\mathbf{k}_1) f_{n'}^*(\mathbf{k}_3 - \mathbf{k}_2 + \mathbf{k}_1) \left(\dot{\boldsymbol{\phi}}_C^{\Lambda,(\ell)}\right)_{\mathrm{C}}^{n,n'}(\nu_3 - \nu_2, \nu_1, \nu_3 - \nu_2 + \nu_1, \mathbf{k}_3 - \mathbf{k}_2) \Big] \ ,
\end{aligned} \tag{49}$$

and $\delta S^\Lambda(p) = G^\Lambda(p) \delta \dot{\Sigma}_1^\Lambda(p) G^\Lambda(p)$.

# 4 Numerical results

In this section we show fRG numerical results obtained with the formalism and code described in the previous sections. After introducing our test system, namely the 2D Hubbard model at half filling, we will test our full momentum-frequency resolved fRG implementation, together

Table 1: Local and first nearest-neighbor form factors both in momentum and real space presentation. For each calculation we specify which form factors are used. A pure $s$-wave calculation restricts to the first line corresponding to the local form factor, the $d$-wave accounts for the first two nearest neighbors form factors, and a calculation with all nearest neighbors form factors includes all five form factors shown here.

|  | $n$ | $f_n(\mathbf{k})$ | $f_n(r_i, r_j)$ |
|---|---|---|---|
| *loc* | 0 | $\frac{1}{2\pi}$ | $\delta_{j,i}$ |
| *1NN* | 1 | $\frac{1}{\sqrt{2}\pi}\cos(k_x)$ | $\frac{1}{\sqrt{2}}(\delta_{j,i+x} + \delta_{j,i-x})$ |
|  | 2 | $\frac{1}{\sqrt{2}\pi}\cos(k_y)$ | $\frac{1}{\sqrt{2}}(\delta_{j,i+y} + \delta_{j,i-y})$ |
|  | 3 | $\frac{1}{\sqrt{2}\pi}\sin(k_x)$ | $\frac{i}{\sqrt{2}}(\delta_{j,i+x} - \delta_{j,i-x})$ |
|  | 4 | $\frac{1}{\sqrt{2}\pi}\sin(k_y)$ | $\frac{i}{\sqrt{2}}(\delta_{j,i+y} - \delta_{j,i-y})$ |

with the inclusion of the self-energy feedback, and study the effect of including multiloop corrections to the $1\ell$ approximated flow equations. If not specified differently, we will make use of a "smooth" frequency-dependent regulator throughout this work:

$$G_0^\Lambda(k) = \frac{\nu^2}{\nu^2 + \Lambda^2} G_0(k), \qquad (50)$$

where $G_0$ specifies the non-interacting Green's function of the 2D Hubbard model. The fRG scheme associated to such a regulator is referred to as $\Omega$-flow [29]. For details on the numerical effort, we refer to the Appendix G.

## 4.1 2D Hubbard model at half filling as test system

As test model we consider the single-band two-dimensional (2D) Hubbard model on the square lattice. Its Hamiltonian reads

$$\hat{\mathcal{H}} = -t \sum_{\langle ij \rangle, \sigma} \hat{c}_{i\sigma}^\dagger \hat{c}_{j\sigma} + U \sum_i \hat{n}_{i\uparrow} \hat{n}_{i\downarrow} - \mu \sum_{i,\sigma} \hat{n}_{i\sigma}, \qquad (51)$$

where $\hat{c}_{i\sigma}^{(\dagger)}$ annihilates (creates) an electron with spin $\sigma$ at the lattice site $\mathbf{R}_i$ ($\hat{n}_{i\sigma} = \hat{c}_{i\sigma}^\dagger \hat{c}_{i\sigma}$), $t$ is the hopping amplitude for electrons between neighboring sites, $\mu$ the chemical potential and $U > 0$ the repulsive on-site Coulomb interaction. In the present study, we consider $U = 2t$, $\mu = U/2$, and different temperature regimes. Since the present model has been extensively studied in the theoretical literature (see, e.g., Refs. [11, 14, 15, 46, 68–71]) as well as in fRG (for a review, see Ref. [1]), it constitutes a reference system to test our novel fRG implementation. Furthermore, the 2D Hubbard model constitutes a delicate case in the context of the Mermin-Wagner theorem [72], which prevents the onset of the antiferromagnetic ordering at finite temperature. Whereas the $1\ell$ fRG results exhibit a pseudocritical Néel temperature $T_{\text{pc}}$, the inclusion of the multiloop corrections to the standard fRG flow should, from a theoretical perspective, recover the parquet solution, which is known to fulfill the Mermin-Wagner theorem [73]. Therefore, we expect $T_{\text{pc}}$ to be suppressed down to 0 in the (converged) multiloop fRG scheme. Despite the rich phase diagram of the 2D Hubbard model out of particle-hole symmetry, we restrict this study to the half-filled particle-hole symmetric case, in order to reduce the numerical efforts.

Let us stress that the bosonic momentum discretization of the first Brillouin zone (BZ) has been chosen such that one obtains a uniform grid along the $x$- and $y$- directions. This represents, though, not the unique choice of resolving the reciprocal space and one could adopt

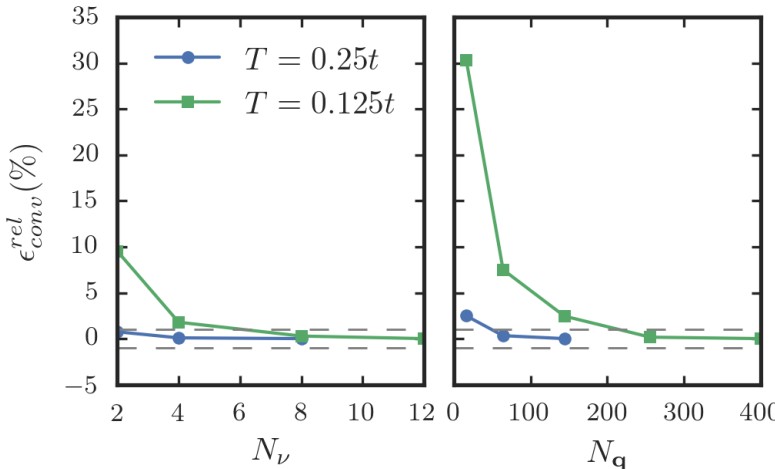

Figure 4: Relative error $\epsilon_{conv}^{rel} = -(\chi - \chi_{conv})/\chi_{conv}$ of the ($1\ell$) AF susceptibilities as a function of the number of fermionic frequencies $N_\nu$ (left) and the number of bosonic momentum patching points $N_{\mathbf{q}}$ (right), for $U = 2t$ and different values of $T$. All calculations are performed with only local ($s$-wave) form factors. In the left panel, $N_{\mathbf{q}}$=144 and $N_{\text{FFT}} = 24 \times 24 = 576$ momentum patching points for the fast Fourier transform. In the right panel, $N_{\mathbf{R}_{\text{FFT}}} = \max(576, 4 \times N_{\mathbf{q}})$ and $N_\nu = 4$ for $T = 0.25$ and $N_\nu = 8$ for $T = 0.125$ respectively. The dashed line corresponds to our tolerance limit of 1%.

some sophisticated "patching" schemes [44], which should be accounted in future optimization of our code.

## 4.2 Convergence and stability study on the TUfRG-implementation

In the previous section 3.1, we presented an efficient parameterization of the vertex which combines the TUfRG scheme [31] for treating momenta with the dynamical fRG implementation proposed in Ref. [34]. In order to illustrate its efficiency of such merge, we have performed a convergence study of the (dominant) antiferromagnetic (AF) susceptibility $\chi_{\text{AF}} = \chi_m^{00}(i\omega_l = 0, \mathbf{q} = (\pi, \pi))$ by means of Eq. (36f), as a function of the number of Matsubara frequencies, momenta and form factors, used in our algorithm. The convergence tests have been performed at temperatures $T = 0.25t \gg T_{\text{pc}}$ and $T = 0.125t \sim T_{\text{pc}}$.

Let us first consider the convergence in the number of fermionic frequencies $N_\nu$ at which the low-frequency structure of the rest-function $\mathcal{R}$ is captured. For $T = 0.25t$ in Fig. 4 (left panel) one observes that the susceptibility does not exhibit significant changes as a function of $N_\nu$. In fact, it is known that, in weakly correlated electron systems, the frequency dependence of the vertex is less important because of power counting arguments [58, 74] and as shown by numerics for small numbers of fermionic Matsubara frequencies, e.g., in Ref. [42]. At $T = 0.125t$ the convergence with respect to $N_\nu$ is slower. According to our tolerance of 1% we obtain convergence at $N_\nu = 8$. In the right panel, we analyze the dependence of the AF susceptibility on the number of bosonic patching points, $N_{\mathbf{q}}$, as shown in Fig. 4. The data for $T = 0.25t$ are already converged at $N_{\mathbf{q}} = 64$, while for $T = 0.125t$ we need $N_{\mathbf{q}} = 256$. In the latter case, one sees that the convergence is more sensitive to $N_{\mathbf{q}}$ than to $N_\nu$. This can be ascribed to the presence of a finite pseudocritical temperature since for $T \to T_{\text{pc}}$ the AF fluctuations become long-ranged, requiring an increasingly finer momentum resolution. At the same time, the size of the objects to handle grows only linearly with $N_{\mathbf{q}}$ while it is expected
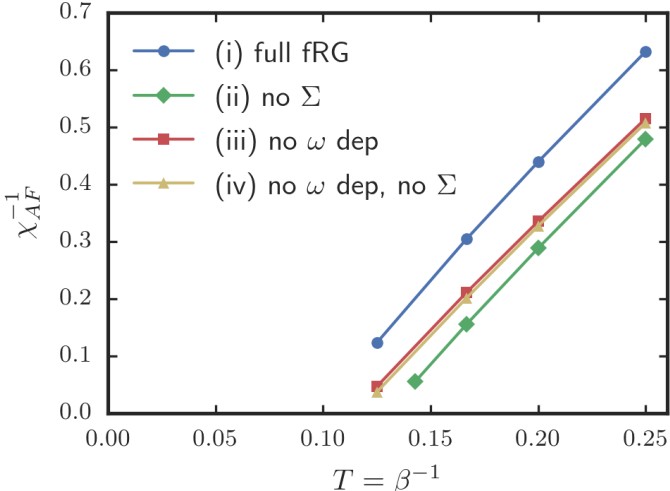

Figure 5: Inverse (1ℓ) AF susceptibility at $\mathbf{q} = (\pi, \pi)$ as a function of temperature, for $U = 2t$. Only local ($s$-wave) form factors are used, but including the nearest-neighbor form factors does not change the results within the accuracy. Besides the curve obtained using the full TU dynamical fRG scheme (i) (blue dots, "full fRG"), different approximations are shown: approximation (ii) (green diamonds, "no $\Sigma$"), (iii) (red squares, "no $\omega$ dep") and (iv) (yellow triangles, "no $\omega$ dep, no $\Sigma$").

to scale with the third power in $N_\nu$, depending on the quantity considered (see Fig. 3). Moreover, the number of independent momentum patching points can be substantially reduced by exploiting point-group symmetries of the lattice.

Last but not least, we have also verified that, for all values of $T$ considered, the AF response function is fully converged with respect to the number of form factors (not shown).

## 4.3 Effects of different approximations

In our fRG scheme, we can choose different approximation levels regarding the treatment of the frequency dependence of the interactions and the self-energy. This allows us to gain a better understanding of the interplay of the different interaction channels and the role of the self-energy.

Here we define four approximation levels (i) to (iv) with decreasing rigor. Approximation (i) represents the fRG treatment described in Sec. (3) which merges the TUfRG scheme with an efficient inclusion of the vertex dynamics; (ii) denotes the flow with a frequency-dependent effective interaction but without the flow and feedback of the self-energy; (iii) is the frequency-independent (static) approximation for the effective interaction and the self-energy, in which the fermion-fermion, fermion-boson and boson-boson vertices are approximated by their value at zero frequency; and (iv) combines the neglect of the self-energy feedback with a static approximation for the vertices.

Approximation (iv) has been the standard one adopted in many previous works, as those reviewed in Ref. [1]. Various other fRG works have already explored the changes occurring by using better approximations like (i) to (iii) introduced above. Earlier studies of the self-energy without explicit frequency dependence of the effective interaction pointed to the possibility of non-Fermi liquid behavior [39, 75]. Later, channel-decomposed fRG [29, 41] and $N$-patch fRG [42] were used to explore the effects of a frequency-dependent effective interaction and of the self-energy feedback. In the following, we rediscover some of their findings, with a

more refined momentum- and frequency-dependent self-energy. Eberlein [43] used a channel-decomposed description of the interaction where each exchange propagator was allowed to depend on one bosonic frequency. He found that in the presence of antiferromagnetic hot spots on the Fermi surface, antiferromagnetic fluctuations lead to a flattening of the Fermi surface and increase the critical scales. Most recently, Vilardi et al. [44] presented a refined $1\ell$ study of the role of the various frequency structures in the interaction, parametrized by three frequencies, albeit with a reduced set of form factors. They argued that a one-frequency parametrization can in some cases lead to spurious instabilities. Our study differs from this work by the ability of taking into account more form factors, using a more economic description of the higher frequencies, and by implementing the multiloop corrections.

In Fig. 5 we show how differently the approximations affect the results for the AF susceptibility. More precisely, we plot the inverse AF susceptibility which decreases quite linearly, i.e., Curie-Weiss-like, upon lowering $T$. The intersection of the curve with the abscissa marks the pseudocritical temperature which, violating the Mermin-Wagner theorem, assumes a finite value in the $1\ell$ fRG scheme. One can observe that the full TU-dynamic fRG approach (i) leads to larger inverse AF susceptibilities, or smaller $\chi_{AF}$, than the other three approximations, shifting $T_{\text{pc}}$ to a smaller value.

Let us first compare the full calculation (i) with the calculation without self-energy but frequency-dependent interactions (ii). It is to be expected that the self-energy renormalizes the leading vertices and therefore also susceptibilities, as has also been observed in fRG studies [29,44]. This explains why the calculations without self-energy flow diverge at higher $T_{\text{pc}}$ with respect to scheme (i).

The flow variants with static interactions (iii) and (iv) differ only slightly. Compared to the fRG flow using scheme (ii), the AF tendencies in these static flows are somewhat weaker as their suppression by particle-particle processes increases when the pairing channel is approximated by its static part, for which it assumes the maximum value. The downward-shift in the inverse AF susceptibility from (iii) and (iv) to (ii) with the inclusion of the frequency dependence of the couplings is however overcompensated by the inclusion of the dynamical self-energy in (i).

Finally, we consider the pseudocritical temperature and the AF susceptibility for the combined approximation of no self-energy and no frequency dependence (iv). Without the screening effect of the self-energy, the pseudocritical temperature increases a little bit more with respect to the static approximation (iii). This has been already observed in Ref. [42]. The small difference may come from the real part of the self-energy that can be understood as upward-renormalization of the hopping parameter, or equivalently a downward-renormalization of the density of states. This is consistent with the self-energy shown below in Fig. 7. For a detailed discussion on the pseudocritical temperatures on a wider range of parameters, we refer the reader to Ref. [41].

## 4.4 Computation of the self-energy

As already implied above, the implementation presented in Sec. 3.1 allows one to compute a frequency and momentum dependence of self-energy during the flow according to Eq. (36a). In Figs. 6 and 7, we present the results for the frequency- and momentum-dependence of the self-energy for different temperatures and momentum points. For the fermionic momentum patching we use the same momentum grid as for the bosonic transfer momentum of the vertex. In the results shown in Fig. 7 (left panel), we subtracted the Hartree contribution, which represents a rigid $U/2$ energy shift at half filling. By looking at Fig. 6, we notice that the frequency dependence of the imaginary part of the self-energy is consistent with a Fermi-liquid, yet without any remarkable difference at different temperatures. As the slope of these curves determines the quasiparticle weight $Z$, we arrive at the conclusion that $Z$ does not decrease

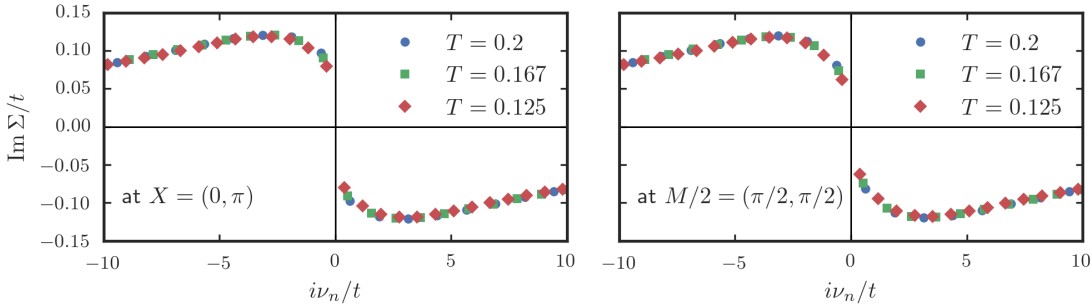

Figure 6: Imaginary part of the self-energy as a function of the Matsubara frequency, at X=$(0,\pi)$ and M/2=$(\pi/2,\pi/2)$, for $U = 2t$ and different temperatures $T = 0.2t$ (blue dots), $T = 0.167t$ (green squares) and $T = 0.125t$ (red diamonds).

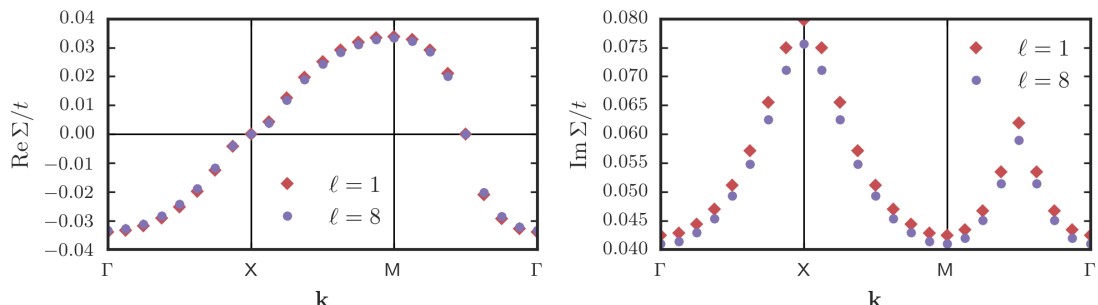

Figure 7: Real (right) and imaginary (left) part of the self-energy $\Sigma(-i\pi T)$ as a function of the bosonic transfer momentum in the $1\ell$ and $8\ell$ truncation of the flow equations, for $U = 2t$ and $T = 0.125t$.

steeply when we lower $T$ towards the AF pseudocritical temperature, as already observed in Refs. [42,43]. Figure 7 shows the momentum dependence of the real and imaginary part of the self-energy along a path in the first BZ defined by $\Gamma = (0,0)$, $X = (0,\pi)$ and $M = (\pi,\pi)$. The fermionic frequency is set to the first fermionic Matsubara frequency. The real part is positive at $M$ and negative at $\Gamma$, while at $X$ and $Y$ it is zero. At lowest order, this momentum structure can be approximated by a positive nearest-neighbor hopping renormalization, which increases the bandwidth. The vanishing of the Fermi surface shift is caused by the particle-hole symmetry of the model at half filling. As for the 2D Hubbard model at half filling, the particle-hole symmetry manifests itself through

$$\Sigma(i\nu,\mathbf{k})^* = -\Sigma(i\nu,(\pi,\pi)-\mathbf{k}), \tag{52}$$

the real part of the self-energy vanishes always at the Fermi surface and the perfect nesting remains intact. This symmetry is not violated by any of the perturbative corrections and also not by the numerical implementation (e.g. the choice of k-points in the BZ). Besides this, there is a substantial bandwidth renormalization that however also reflects the symmetries of the system, i.e. it has opposite sign at $\Gamma$ and at M. The $8\ell$ results in Fig. 7 will be discussd in Sec. 4.5.

The imaginary part of the self-energy shows two peaks around X and M/2= $(\pi/2,\pi/2)$. This corresponds to a maximal scattering on the nested Fermi surface and minimal on the points $\Gamma$ and M, which are at maximal distance from the Fermi surface. Note that this refers to the self-energy at small fixed imaginary frequency and not at real frequency equal to the

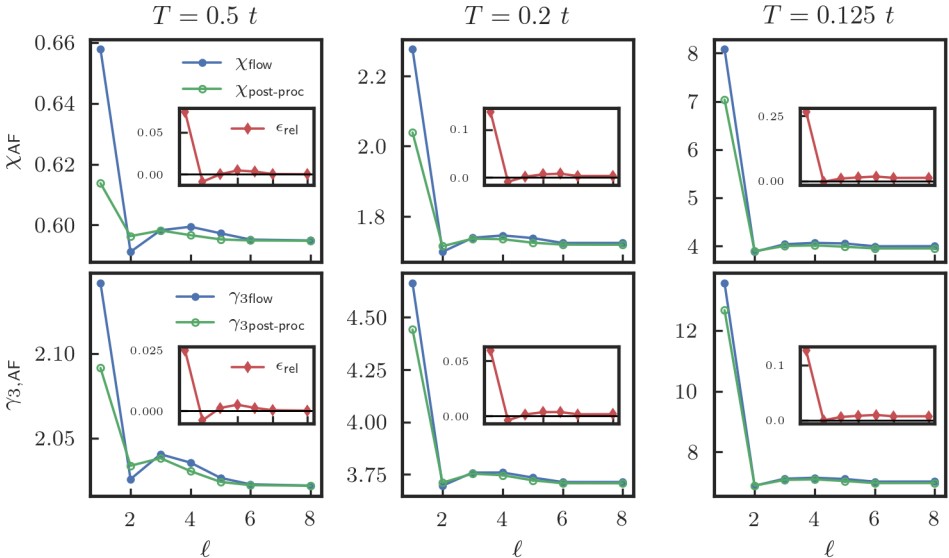

Figure 8: AF susceptibility (upper panels) and fermion-boson vertex (lower panel) at $\mathbf{q} = (\pi, \pi)$ as a function of the number of loops, for $U = 2t$ and $T = 0.5t, 0.2t, 0.125t$ (from left to right). The susceptibility is evaluated at $\omega = 0$ and the fermion-boson vertex at $\omega = 0$ and $\nu = \pi/\beta$. The blue line shows the behavior of the integrated Eq. (23a) up to $\ell = 8$, while the green line the one obtained from the post-processed calculation by means of Eq. (8) for $\chi$ and of (10) for $\gamma_3$. The insets show the relative difference between the blue and the green lines, defined for the susceptibility as $\epsilon_{\text{rel}} = (\chi_{\text{flow}}^{\ell} - \chi_{\text{post-proc}}^{\ell})/\chi_{\text{post-proc}}^{\ell=8}$.

excitation energy, i.e., this behavior does not contradict the typical behavior that the scattering rates for quasiparticles rise with distance from the Fermi surface.

## 4.5 Effect of the multiloop implementation

Let us now investigate the effect of including multiloop corrections to the flow equations of the susceptibility and the fermion-boson vertex as in Eq. (23a). As previously discussed, the inclusion of the multiloop corrections should allow us to recover the full derivative of Eq. (8) and (10) with respect to the scale parameter $\Lambda$. This means that the integration of the multiloop fRG flow equations should converge, by increasing the number of loops, to Eq. (8) and (10), as well as to the parquet equations for $\gamma_4$ and $\Sigma$ as discussed in Ref. [55].

Although, in the half-filled case, the numerical effort is already reduced compared to the non-particle-hole symmetric situation, calculations for $T < 0.5t$ are already quite demanding if a multiloop cycle is included. Therefore, the only calculations involving more than one form factor (i.e., $s$-wave) that will be presented here were performed at a rather high temperature of $T = 0.5t$. Despite this restriction, since the physics of the single band Hubbard model at half filling is dominated by the AF fluctuations, the fRG results are already converged in the number of form factors. Nevertheless, a meaningful part of the $d$-wave susceptibilities is still accessible, as it will be shown in the following, via the $s$-wave two-particle vertex.

In Fig. 8 we show the s-wave susceptibility $\chi$ (fermion-boson vertex $\gamma_3$) in the upper (lower) panels in the magnetic channel for $i\omega_l = 0$ ($i\omega_l = 0$ and $i\nu_o = \pi/\beta$ for $\gamma_3$) and $\mathbf{q} = (\pi, \pi)$ as a function of the number of loops considered in the mfRG calculation, for three

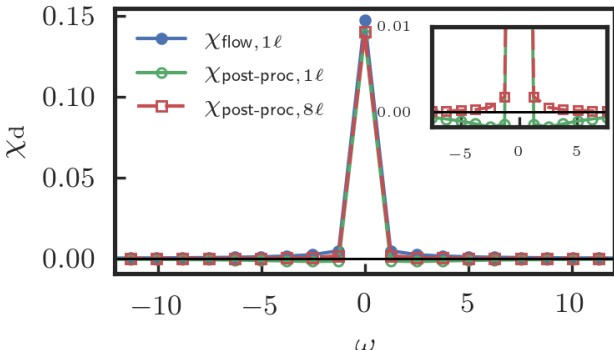

Figure 9: $S$-wave density susceptibility evaluated at $\mathbf{q} = (0,0)$ as a function of bosonic Matsubara frequencies, for $U = 2t$ and $T = 0.2t$. The blue and green lines represent the flow and post-processed values for $1\ell$, while the red dashed line corresponds to the post-processed mfRG result for $8\ell$. The zoom in the inset shows that the post-processed $1\ell$ data assume unphysical negative values at finite frequencies.

selected temperatures $T = \{0.5t, 0.2t, 0.125t\}$ (left to right). The blue lines show the value of $\chi$ and $\gamma_3$ calculated by the integration of Eq. (23a). On the other hand, the green lines show $\chi$ ($\gamma_3$) acquired at the end of the $\ell$-loop fRG flow ($\Lambda = \Lambda_{\text{fin}}$) by means of Eq. (8) ((10)), where we inserted on the r.h.s $\gamma_4^{\Lambda_{\text{fin}}}$ and $G^{\Lambda_{\text{fin}}}$, referred to in Section 2.2 as "post-processed" method. In the present case, one sees how the convergence of the two lines is achieved after $8\ell$ for all temperatures presented. Thus, we have a dual convergence: as a function of the loop number and between two ways of computing the same quantity. Clearly, by decreasing the temperature and approaching the $1\ell$ fRG pseudocritical temperature (see Fig. 11), the antiferromagnetic (AF) susceptibility and $\gamma_{3,\text{m}}^{00}(\omega = 0, \nu = \pi/\beta, \mathbf{q} = (\pi, \pi)) = \gamma_{3,\text{AF}}$ increase and the green and blue lines for the two ways to compute the susceptibility exhibit the largest relative difference at $\ell = 1$ of $\sim 25\%$. This difference decreases by increasing the loop number down to less then 1% for $\ell = 8$.

It is interesting to see the main effect of the multiloop corrections occurs already at the $2\ell$ level, where the $1\ell$ results experience the strongest screening effect. Furthermore, as explicitly argued in Ref. [34] the inclusion of the two-loop corrections to the flow of the interaction allows to substantially enrich the virtual excitation content of the fRG equations. By looking at Fig. 8 one could deduce that, performing a post-processed evaluation of the susceptibility, as well as of the fermion-boson vertex, brings them closer to the converged values than the corresponding results coming from the fRG flow (blue curves). However, it has to be stressed that the convergence trend observed in the magnetic channel for the post-processed $\chi$ and $\gamma_3$ does not apply in general. Counterexamples can be observed, for instance, in the $s$-wave secondary channels (i.e., charge and superconducting), where the post-processed evaluation of the $1\ell$ susceptibility not only leads to an overscreening (i.e., an underestimation with respect to the converged result), but, e.g., in the charge channel, to even unphysical results, as can be observed in Fig. 9. Here, the $s$-wave susceptibility in the density channel is plotted at $\mathbf{q} = (0,0)$ as a function of the bosonic Matsubara frequencies. One observes negative values of the post-processed susceptibility (green line) at finite bosonic frequencies, which are restored to positive values by the multiloop corrections (red line). An attempt to explain this different trend between the dominant (magnetic) and the secondary channels (density and superconducting) is extensively discussed in Appendix D and summarized in the following observations.

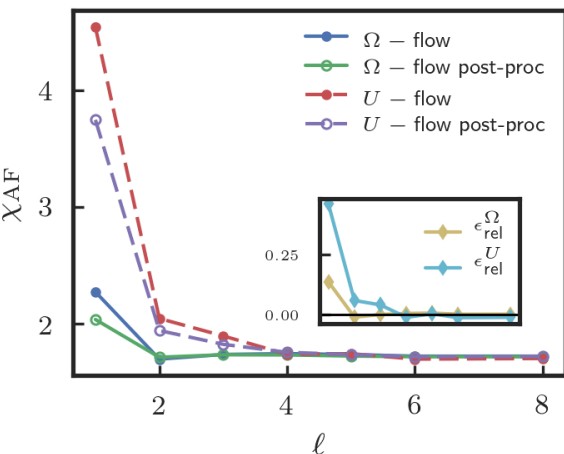

Figure 10: Comparison of two cutoff schemes, the $U$-flow and the $\Omega$-flow, for the AF susceptibility as a function of the number of loops, for $U = 2t$ and $T = 0.2t$. *Inset*: Relative difference with respect to the converged value $(\chi_{AF}^{\ell} - \chi_{AF}^{\text{post-proc},\ell})/\chi_{AF}^{8\ell}$.

As explicitly derived in Appendix D, the $\Lambda$-derivative of the formal definition for the susceptibility reported in Eq. (8) (as well as Eq. (10) for $\gamma_3$), after substituting the derivative of $\gamma_4$ and $\Sigma$ by their $1\ell$ fRG flow equations, leads to additional terms with respect to the standard $1\ell$ flow equations for $\chi$ in Eq. (19a) (for $\gamma_3$ in Eq. (19b)). These terms, besides self-energy derivative corrections (which are generally introduced starting from the second loop-order under the name of *Katanin* corrections [53]), have a $3\ell$-like topological diagrammatic form (see Eq. (26)). The internal loops of $\mathbf{i}_\eta^{\Lambda,(1)}$ (marked in red in Fig. 16) contained in such terms act as a screening effect provided by the complementary channels ($\eta'$) to the one considered ($\eta \neq \eta'$). Because of the imbalance between the $1\ell$ approximation for the two-particle vertex $\gamma_4$ and $\Sigma$, and the $3\ell$ diagrams included in the modified "post-processed flow equation" for the susceptibility (see Appendix D), this screening effect ends up being overestimated. Nonetheless, it represents a minor effect on the dominant (magnetic) channel, where the imbalance effect is still governed by the large $1\ell$ antiferromagnetic contribution. It could however lead to major changes in the secondary channels, which are affected by the strong screening effect of the magnetic channel appearing on the $3\ell$-like terms. The overscreening affects all frequencies, because of the internally summed diagrams. Therefore, it is particularly severe at nonzero frequencies where the susceptibility assumes small values. This explains the unphysical negative values of the density susceptibility in Fig. 9.

By applying different fRG cutoff schemes, we obtain further tests of the reconstruction of the full derivative of Eq. (8) provided by the multiloop approach. In Fig. 10 we compare the results shown already in Fig. 8 (central upper panel) for $T = 0.2t$ using a frequency-dependent regulator ($\Omega$-flow) with the results for $\chi$ at the same temperature obtained by a trivial or flat regulator, also known as interaction or $U$-cutoff [76]. Differently from the $\Omega$-flow, the $U$-flow just multiplies the bare propagator with a scale factor that is increased from 0 to 1. Hence, it does not provide any cutoff in energy during the fRG flow so that all energy scales are treated on an equal footing. The insertion of the multiloop corrections into the fRG flow equations, as already observed in a different system in Ref. [56], makes the mfRG calculation almost independent, at high enough loop-order, from the specific regulator considered. A more detailed analysis of our results revealed a persisting small discrepancy even for higher loops. Since it vanishes in absence of self-energy corrections, we attribute it to the truncation of the form factor basis in the vertex flow which prevents the reconstruction of the full derivative of the self-energy.

The substantial reduction of the pseudocritical temperature ($T_{pc}$) provided by the multi-loop corrections can be easily inferred from the data in Fig. 11. Here, the inverse $1\ell$ fRG anti-ferromagnetic susceptibility (blue line) is plotted as a function of temperature and compared to the one computed with $8\ell$ mfRG calculation (green line): at any temperature considered the higher-loop corrections systematically suppress the value of the susceptibility, thus lowering the pseudocritical scale.

We note that the formal equivalence between the mfRG and the parquet approximation should guarantee the fulfillment of the Mermin-Wagner theorem [72] as this is fulfilled by the parquet approximation [64]. Hence, a frequency-momentum *and* loop converged mfRG calculation should yield a complete suppression of the pseudocritical temperature down to zero. It is, however, very hard to prove this result by means of direct calculations in the low-$T$ regime, due to the quasi-long-range nature of the spatial fluctuations, responsible for the Mermin-Wagner theorem. In fact, the "avoided" onset of a true long-range antiferromagnetism at finite temperature $T$ is associated with the appearance of antiferromagnetic fluctuations with an exponential growing correlation length (see, e.g, discussion in Ref. [73]). Their occurrence has been indeed explicitly verified in several many-body calculations [9, 10, 73, 77–80] compatible with the Mermin-Wagner theorem. While these low-temperature exponentially extended correlations make the overall physics of our system very similar to that of a true AF ordered phase [81], being associated with a rapid crossover towards a low-temperature insulating behavior, they also make it numerically impossible to access the $T \rightarrow 0$ limit, because of the finiteness of any momentum grid discretization. In fact, in the temperature range where we could achieve a satisfactory momentum-convergence of our $8\ell$ results the antiferromagnetic susceptibility does not show yet any evidence of the exponential behavior expected in the low-temperature regime. On the contrary for almost all the data, one still observes a linear mean-field like behavior for the inverse susceptibility (though significantly renormalized w.r.t. the $1\ell$ results). As a consequence, a reliable low-$T$ extrapolation for estimating $T_{pc}$ from our converged $8\ell$ results is not possible: If trying to extrapolate the data of Fig. 11, one would rather obtain an estimation for the instability scale of an effectively renormalized mean-field description, valid in the high-$T$ regime.

Our findings and considerations are consistent with the most recent estimates of the temperature range, below which the exponential behavior of $\chi_{AF}$ should become visible: According to the most recent DΓA and Dual Fermion studies [9, 10, 15, 80, 82] such a "crossover" temperature would be *lower* than the ordering temperature of DMFT. The latter, for $U = 2$ is $T_N^{DMFT} \sim 0.05$ ($\beta = 20$), i.e., already twice smaller than the lowest temperature considered in the present work. We also observe that this DMFT critical scale would be roughly in agreement with the linear extrapolation of our $8\ell$ data for the inverse susceptibility discussed above.

Next, we analyze the effect of the fRG multiloop corrections on some $d$-wave physical susceptibilities which, although suppressed in the particle-hole symmetric case, play an important role in describing the phase diagram of the 2D Hubbard model, most notably away from half filling [28, 36, 58, 83, 84]. In particular we analyze the static ($\omega = 0$) $d$-wave susceptibility in the superconducting channel for $\mathbf{q} = (0, 0)$ (dSC), as well as the static $d$-wave susceptibility in the charge channel for a bosonic momentum transfer $\mathbf{q} = (0, 0)$ (dPom), which would become dominant in the case of the so-called "Pomeranchuk" instability. The staggered $d$-wave charge density wave (dCDW) susceptibility for $\mathbf{q} = (\pi, \pi)$ has not been shown because of its degeneracy with the correspondent $d$-wave superconducting one. In fact, one can formally demonstrate that in a SU(2) and particle-hole symmetric case, where the system becomes invariant under pseudospin rotation, the pairing susceptibility at $\mathbf{q} = (0, 0)$ associated to a specific symmetry of the order parameter is degenerate with the staggered ($\mathbf{q} = (\pi, \pi)$) CDW associated to that specific symmetry. In Fig. 12 we display the result of a fRG calculation where, in addition to the $s$-wave form factor, the form factors indicated as 1 and 2 in Table 1 have been used. As

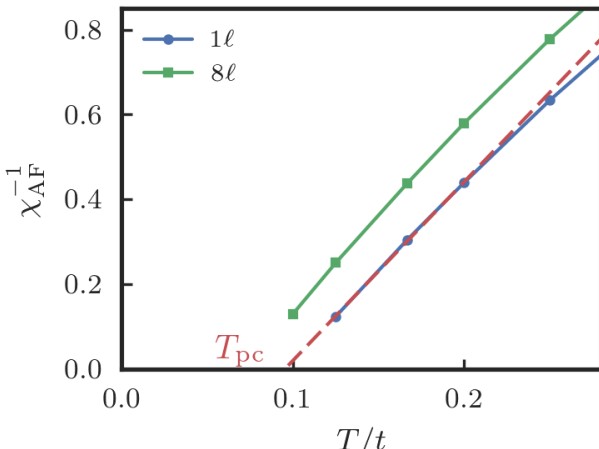

Figure 11: Inverse AF susceptibility as a function of temperature, for $U = 2t$.

in Fig. 8, the blue line indicates the fRG result obtained by the integration of Eq. (23a) up to a specific $\ell$-loop order, alongside the corresponding ($\ell$-loop) mfRG equations for $\Sigma$ and $\gamma_4$. The green line represents the post-processed result for the $d$-wave susceptibilities calculated from a $s+d$-wave $\ell$-loop order mfRG results for the self-energy ($\Sigma$) and the two-particle vertex ($\gamma_4$). The red line has been obtained, similarly to the green one, from $s$-wave $\ell$-order mfRG results for $\Sigma$ and $\gamma_4$. One notices that, differently to the antiferromagnetic case, the relative difference between blue and green lines with respect to the convergence value is, at the $1\ell$-level, of the order of few percents and lowers even down to less then $1\%_{oo}$ at $8\ell$. Interestingly, the post-processed susceptibilities obtained from the $s$-wave fRG results (red curve) are almost on top of the correspondent ones where both $s$- and $d$-wave form factors have been considered during the fRG flow. This shows clearly that, as already known from previous studies on the single-band 2D Hubbard model, the $d$-wave tendencies in pairing and charge channels are triggered by the antiferromagnetic fluctuations of onsite ($s$-wave) spin bilinears. However, according to our data for the Fermi surface and the temperature considered, the flow of $d$-wave pairing and charge channels, which are not captured if only $s$-wave interactions flow, does not seem to be particularly relevant. This means that in the full system where all channels ($s$-wave, $d$-wave, etc.) are allowed to flow, the $d$-wave attractions triggered by the $s$-wave AF fluctuations would not fall on a too fertile ground at $T = 0.5t$, i.e., they would not flow strongly in their 'native' $d$-wave channels. Going to lower $T$ and in particular out of half filling, this will likely change, as the particle-particle diagrams will enhance any attractive pairing component. Therefore, it is a priori not clear if the $d$-wave susceptibilities computed at lower $T$ by projecting the vertex made up from $s$-wave bilinears could provide satisfactory physical results. Nevertheless, we argue that they serve as useful theoretical test objects for the convergence in the order of the multiloop corrections. This is because the effective $d$-wave interactions captured this way can be understood as two-particle irreducible (2PI) interactions in the $d$-wave pairing or charge channels, generated purely by $s$-wave one-loop processes. These 2PI $d$-wave quantities are non-singular but zero at lowest order in $U$ in typical cases. Hence they can be expected to be dominated by diagrams of finite order in $U$ that should exhibit stronger multiloop effects. In contrast with these terms, the missing boosts in the respective native channels, e.g., in the pairing channel, would just be a higher-order ladder summation of, for $T \to 0$, increasingly singular one-loop diagrams. Hence, if multiloop convergence is reached in the two-particle irreducible interactions, it is likely that the same degree of convergence would be found in the true susceptibilities. This idea leads us to consider the data shown in Fig. 13.

As already visible for $T = 0.5t$ in Fig. 12, the post-processing calculations exhibit a weak

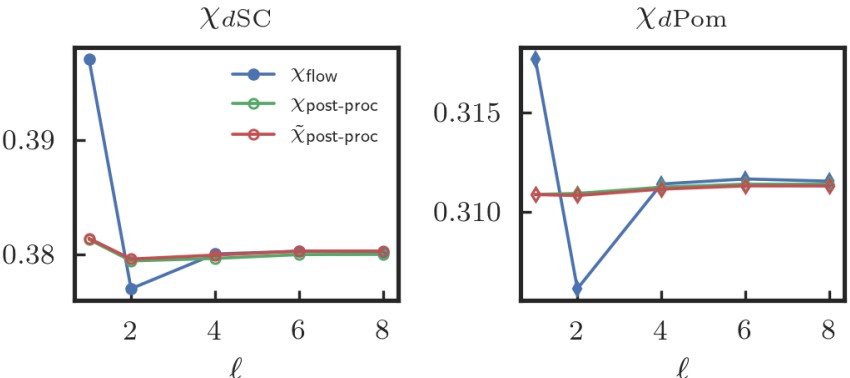

Figure 12: $d$-wave susceptibililties $d$SC, $d$Pomeranchuk ($\mathbf{q} = (0,0)$) at $i\omega_m = 0$ as a function of the number of loops, for $U = 2t$ and $T = 0.5t$. The red line has been evaluated by means of Eq. (8), by inserting the two-particle vertex computed from a single ($s$-wave) form factor.

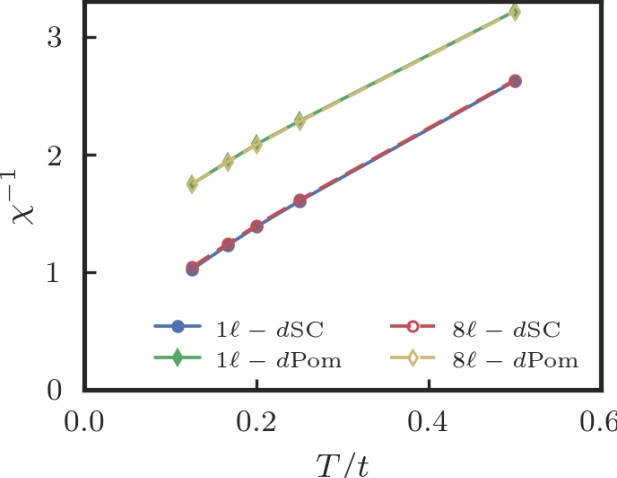

Figure 13: Inverse $d$-wave susceptibilities, computed by post-processing, as a function of temperature, for $U = 2t$ (fRG flow with only $s$-wave bilinear interactions).

dependence on the loop number (with a relative fluctuation less than 1‰). This is confirmed in Fig. 13 where the post-processed inverse $d$-wave susceptibilities in the aforementioned channels are calculated out of an $s$-wave 1$\ell$ (blue and green lines) and 8$\ell$ (red and yellow dashed lines) fRG flow. As it is apparent in the figure, the effects of the multiloop corrections are insignificant compared to the variation of the inverse susceptibilities in temperature.

To conclude this section, we comment on the multiloop effects on the self-energy shown in Fig. 7. The bandwidth renormalization is changed insignificantly and the scattering on the Fermi surface is reduced only slightly. Also the Fermi-surface shift remains zero in mfRG because the particle-hole symmetry is preserved in fRG, in PA and in the full solution and therefore also the multiloop corrections do not violate the particle-hole symmetry.

## 5   Conclusions

We have presented a comprehensive study of forefront algorithmic implementations of the fRG for interacting fermions on 2D lattices. While we focused on the 2D Hubbard model, the

methodological improvements discussed here can provide a useful guidance for the generalization to other systems.

Our main goal is to illustrate the progress achieved when going beyond the approximations routinely made in most previous fRG computations. In particular, we have worked on the following aspects: (i) an accurate and converged treatment of both the momentum and frequency dependence of the vertex function together with its asymptotic structures; (ii) the inclusion of the self-energy and its feedback in the fRG flow; (iii) the implementation of the multiloop corrections beyond the standard $1\ell$.

Regarding the first aspect (i), we have kept the more general dependence of the two-particle vertex on all *three* Matsubara frequencies. We extend previous works [41, 44, 84–87] by exploiting an "economic" description [34] provided by an efficient parametrization of the high frequency asymptotics [32]. We could show that this parametrization can be brought to convergence in the number of frequencies employed, i.e., the results do not change if more frequencies are used. We combined this treatment of the frequency dependence with the truncated-unity technique for the momentum dependence, whose form-factor expansion was also shown to converge quickly for our test case [31].

With a frequency-dependent flowing interaction, we could also compute a momentum- and frequency-dependent self-energy, which has been fed back into the flow of the two-particle vertex. Through a systematic analysis of specific observables – in particular of the response functions – we could assess the effects of the improved algorithmic implementation with respect to previous results and demonstrate how, for the parameters studied, the fRG results can be converged in the number of considered frequencies. An analogous convergence could be also established for the 2D momentum dependence.

The major advancement achieved in this work is, however, the implementation of the multiloop corrections both for the flow of the two-particle vertex as well as for the flow of the coupling to external fields and the corresponding susceptibilities. The multiloop extension, so far only tested for a (prototypical) toy model [56], adds more virtual excitations to the flow of the two-particle vertex compared to the previously used $1\ell$ truncation. As it was diagrammatically shown [55, 56], if truncated fRG results are converged with respect to the loop order, they exactly reproduce the parquet approximation (PA), not only concerning the topology of the summed diagrams, but also – quantitatively – their precise weight. This has been also recently confirmed by a formal analytical derivation of the multiloop fRG equations [57]. From this property, it follows that the results of a loop-converged fRG algorithm become completely independent from the employed cutoff scheme, at least if all modes are integrated out at sufficiently high temperature.

Previously, it was not clear how the contributions missing in the $1\ell$ truncation would influence the results quantitatively. On the numerical level, the effort for including the multiloop corrections to order $\ell$ only rises linearly in $\ell$, i.e. the situation is far better than if one really had to compute all higher-loop diagrams. Our studies show that the multiloop corrections can be included also in 2D up to rather high orders of $\ell = 8$. We find that the observables converge quite nicely when the multiloop order is increased. While it is not obvious that this quick convergence will hold for all model parameters and for all models of interest, our study shows that these checks can be performed with feasible numerical efforts. This adds a new important degree of quantitative control to the fRG, at least in the weak to intermediate coupling regime where the PA can be considered accurate. At stronger coupling, where low-frequency vertex corrections beyond the PA might appear [32, 50, 88–90], the mfRG could provide a much better [15] setup for the proposed combination with the DMFT [23, 27, 91]. The loop convergence of our fRG results is also reflected in the progressive reduction of the dependence of our fRG results on the chosen cutoff scheme, which appears completely suppressed at the $8\ell$ level.

The incorporation of the multiloop contributions has also another rather appealing and

quantitatively important aspect, giving rise to an additional very useful type of convergence. It has been known that response functions can be computed in two different ways in RG approaches and that the results differ due to the involved approximations. One way is to consider the flow of couplings of 'composite operator' bilinears in the primary degrees of freedom to external fields of appropriate type. Then the response function is obtained as renormalization of the propagator of the external field.

The other way, referred to as post-processing, is to compute the response functions by means of their diagrammatic expression, evaluated from the dressed bare fermion bubbles and the two-particle vertex at the end of the flow. In fact, in some cases arguments were made (see, e.g., Ref. [92] and references therein) that the external field methods should give more controlled results, i.e., that composite operators should be renormalized separately, because, at the level of the approximations made, the post-processed quantities, which involve the integration over all energies and momenta, are more affected by approximation errors. In our study, the multiloop extension of the response function flow allows us to show that *also* the flow of the response functions becomes an exact scale derivative of the post-processed response function. This establishes the formal equivalence of the two ways to compute response functions on the multiloop level. This formal equivalence is remarkably reflected by our numerical results, which exhibit a clear convergence of the two approaches: If the multiloop convergence is achieved, and frequency and momentum dependencies as well as the self-energy feedback are included appropriately, the fRG results for the response functions are unambiguous. The corresponding data can be used for quantitative studies and directly compared with other numerical techniques or with experiments, if the effective modelling of the problem is sufficiently realistic.

In summary, our study shows how the fRG algorithms for two-dimensional fermionic lattice models can be brought to a quantitatively reliable level at weak to moderate couplings, as long as the parquet approximation is appropriate. This goal has been reached by means of an economic, but accurate, treatment of the momentum and frequency dependencies which takes into account the asymptotic structure of the two-particle vertex and the self-energy during the fRG flow. This fRG framework has been supplemented with the implementation of the multiloop corrections to the $1\ell$ truncation scheme.

The current work concentrates on testing the improved fRG method in a situation that is reasonably well understood. The fRG method itself is however not limited to this situation and can be applied to situation where the landscape of instabilities and emergent energy scales is less explored. For instance, within the framework of the 2D Hubbard model, we could apply our algorithmic implementation to broader parameter regimes in future works. If the Fermi surface displays a given curvature, due to the inclusion of, e.g., more hopping terms or changes of the band filling, the dominance of the AF channel will be weakened and the pseudo-critical scales will become smaller. For such cases the convergences of the different approximation might possibly vary. In particular, since the generation of $d$-wave pairing tendencies in third order of the bare coupling involves $2\ell$ diagrams that are only partially captured in the $1\ell$ truncation, we would expect the impact of the multiloop corrections to become more noticeable.

## Acknowledgements

The authors thank A. Eberlein, J. Ehrlich, P. Hansmann, J. Lichtenstein, W. Metzner, T. Reckling, D. Sánchez de la Peña, G. Schober, C. Taranto, J. von Delft, D. Vilardi, and N. Wentzell for valuable discussions, D. Rohe (FZ Jülich) for major help on the parallelization of the code, and W. Metzner for a careful reading of the manuscript.

**Author information**    A. Tagliavini and C. Hille contributed equally to this work.

**Funding information**    We acknowledge financial support from the Deutsche Forschungsgemeinschaft (DFG) through ZUK 63, RTG 1995, HO 2422/11-1, and Projects No. AN 815/5-1 and No. AN 815/6-1, and the Austrian Science Fund (FWF) within the Project F41 (SFB ViCoM) and I2794-N35. Calculations were performed on the Vienna Scientific Cluster (VSC), at Jülich Supercomputing Centre (JSC) in the frame of project jjsc28, and at the MPI for Solid State Research in Stuttgart.

# A    Symmetries and symmetrized notation

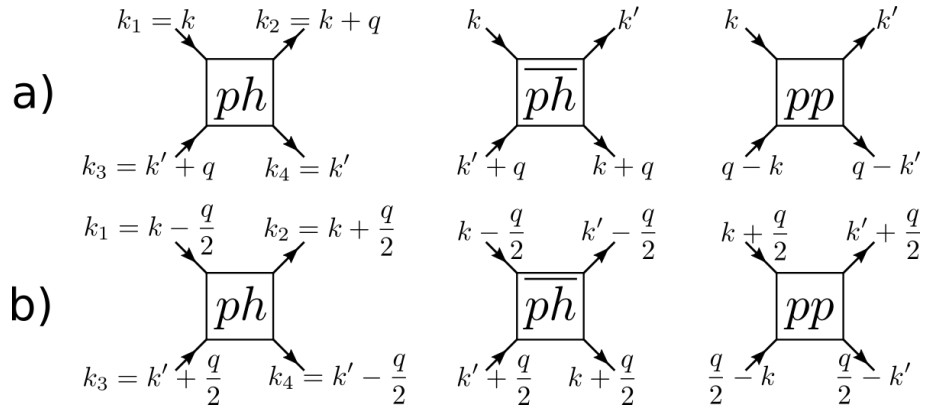

Figure 14: (a) Non-symmetrized and (b) symmetrized notation for the vertices, reducible vertices and irreducible vertices in the diagrammatic channel notation. The non-symmetrized notation is used primarily here, while the symmetrized notation is used only in App. A.

Here we illustrate how diagrammatic and lattice related symmetries can be expressed in an easy way and how they are implemented in our code. Directly related to the symmetries is the question if one uses the symmetrized or the non-symmetrized notation illustrated in Fig. 14 for the momentum and frequency dependence of the channels. In Section 3.1 we argued that the non-symmetrized notation leads to more readable flow equations, bubbles and projection matrices. Therefore we adopted primarily this notation. The symmetries, however, are much easier to express in the symmetrized notation. While in the non-symmetrized notation, simple relations like the crossing relation involve multiple form factor combinations, in the symmetrized notation we find a one-to-one correspondence. Therefore we here use for both momentum and frequency the symmetrized notation ($s$), which is related to the non-symmetrized ($ns$) by

$$\phi_{ph}^{s}(q,k,k') = \phi_{ph}^{ns}\left(q, k-\frac{q}{2}, k'-\frac{q}{2}\right) \tag{53a}$$

$$\phi_{\overline{ph}}^{s}(q,k,k') = \phi_{\overline{ph}}^{ns}\left(q, k-\frac{q}{2}, k'-\frac{q}{2}\right) \tag{53b}$$

$$\phi_{pp}^{s}(q,k,k') = \phi_{pp}^{ns}\left(q, k+\frac{q}{2}, k'+\frac{q}{2}\right). \tag{53c}$$

## A.1    Lattice related symmetries

First we specify how lattice related symmetries are reflected in the form factor expansion of the channels in the symmetrized notation. The lattice symmetries always depend on the system

and we here focus on the 2D Hubbard model on a square lattice, where we have for example the rotation of $\pi/2$ around the $z$-axis and the mirroring at the $y$-axis as independent symmetry operations. Under any of these operations, or combinations of them, applied simultaneously to all momentum dependencies, the expressions of the channels are invariant. This can be translated into the form factor expansion by

$$
\begin{aligned}
\hat{P}[F]_{n,n'}(\mathbf{q}) &= \int d\mathbf{k}d\mathbf{k}'f_n^*(\mathbf{k})f_{n'}(\mathbf{k}')F(\mathbf{q},\mathbf{k},\mathbf{k}') \\
&= \int d\mathbf{k}d\mathbf{k}'f_n^*(\mathbf{k})f_{n'}(\mathbf{k}')F(\hat{R}(\mathbf{q}),\hat{R}(\mathbf{k}),\hat{R}(\mathbf{k}')) \\
&= \int d\mathbf{k}d\mathbf{k}'f_n^*(\hat{R}^{-1}(\mathbf{k}))f_{n'}(\hat{R}^{-1}(\mathbf{k}'))F(\hat{R}(\mathbf{q}),\mathbf{k},\mathbf{k}')\,,
\end{aligned}
\tag{54}
$$

where $F$ is any of the channels $D$, $C$ or $P$. The frequency dependence is not affected and is therefore omitted. We here exploited the symmetry under consideration and introduced a variable change. If the form factors are chosen in such a way that under this symmetry operation any form factor is related to a linear combination of others, described by the matrix $\mathbf{M}_{\hat{R}^{-1}}(\mathbf{k})$, it holds in addition

$$
\hat{P}[F]_{n,n'}(\mathbf{q}) = \int d\mathbf{k}d\mathbf{k}'\sum_m f_m^*(\mathbf{k})M_{\hat{R}^{-1}}(\mathbf{k})_{mn}\sum_{m'}M_{\hat{R}^{-1}}(\mathbf{k}')_{n'm'}f_{m'}(\mathbf{k}')F(\hat{R}(\mathbf{q}),\mathbf{k},\mathbf{k}')\,.
\tag{55}
$$

If moreover, the symmetry operation on every form factor yields a single other form factor expressed by the vector $\mathbf{V}_{\hat{R}^{-1}}$, the above relation simplifies to

$$
\hat{P}[F]_{n,n'}(\mathbf{q}) = \hat{P}[F]_{V_{\hat{R}^{-1}(n)}V_{\hat{R}^{-1}(n')}}(\mathbf{q})S_{V_R(n)}S_{V_R(n')}\,,
\tag{56}
$$

where the only difference is a possible sign change taken into account by $S_{V_R(n)}$. These assumptions hold for the form factors used in the present implementation (see Table 1), but are not necessarily valid for an arbitrary choice of form factors.

## A.2  Diagrammatic symmetries

In addition to the lattice related symmetries, there are diagrammatic symmetries which are independent of the geometry of the system. Considering a two-particle fermionic vertex, we can apply the crossing symmetry simultaneously to the annihilation and the creation operators, recovering the following relations:

$$
F_{\sigma_1,\sigma_2,\sigma_3}(i\nu_{o_1},i\nu_{o_2},i\nu_{o_3},k_1,k_2,k_3) = F_{\sigma_3,\sigma_4,\sigma_1}(i\nu_{o_3},i\nu_{o_4},i\nu_{o_1},k_3,k_4,k_1)\,,
\tag{57}
$$

time reversal

$$
F_{\sigma_1,\sigma_2,\sigma_3}(i\nu_{o_1},i\nu_{o_2},i\nu_{o_3},k_1,k_2,k_3) = F_{\sigma_2,\sigma_1,\sigma_4}(i\nu_{o_2},i\nu_{o_1},i\nu_{o_4},k_2,k_1,k_4)\,,
\tag{58}
$$

and complex conjugation

$$
F_{\sigma_1,\sigma_2,\sigma_3}^*(i\nu_{o_1},i\nu_{o_2},i\nu_{o_3},k_1,k_2,k_3) = F_{\sigma_2,\sigma_1,\sigma_4}(-i\nu_{o_2},-i\nu_{o_1},-i\nu_{o_4},k_2,k_1,k_4)\,,
\tag{59}
$$

for which we refer to Ref. [93]. In the SU(2) symmetric case, by projecting the vertex $\phi$ to the form factor basis and adopting the symmetrized notation, one has that Eq. (57) gives

$$
P_{n,n'}(i\omega_m,i\nu_o,i\nu_{o'},\mathbf{q}) = \Pi_n\Pi_{n'}P_{n,n'}(i\omega_m,-i\nu_o,-i\nu_{o'},\mathbf{q})
\tag{60a}
$$

$$
D_{n,n'}(i\omega_m,i\nu_o,i\nu_{o'},\mathbf{q}) = D_{n,n'}(-i\omega_m,i\nu_{o'},i\nu_o,-\mathbf{q})
\tag{60b}
$$

$$
C_{n,n'}(i\omega_m,i\nu_o,i\nu_{o'},\mathbf{q}) = C_{n,n'}(-i\omega_m,i\nu_{o'},i\nu_o,-\mathbf{q})\,,
\tag{60c}
$$

where $\Pi_m$ is the parity associated to the momentum inversion of the form factor $m$ defined as

$$f_n(-\mathbf{k}) = \Pi_n f_n(\mathbf{k}) \, . \tag{61}$$

The time reversal symmetry reads

$$P_{n,n'}(i\omega_m, i\nu_o, i\nu_{o'}, \mathbf{q}) = P_{n',n}(i\omega_m, i\nu_{o'}, i\nu_o, \mathbf{q}) \tag{62a}$$

$$D_{n,n'}(i\omega_m, i\nu_o, i\nu_{o'}, \mathbf{q}) = D_{n,n'}(-i\omega_m, i\nu_o, i\nu_{o'}, -\mathbf{q}) \tag{62b}$$

$$C_{n,n'}(i\omega_m, i\nu_o, i\nu_{o'}, \mathbf{q}) = C_{n',n}(i\omega_m, i\nu_{o'}, i\nu_o, \mathbf{q}) \, , \tag{62c}$$

and the complex conjugation

$$P^*_{n,n'}(i\omega_m, i\nu_o, i\nu_{o'}, \mathbf{q}) = P_{n',n}(-i\omega_m, -i\nu_{o'}, -i\nu_o, \mathbf{q}) \tag{63a}$$

$$D^*_{n,n'}(i\omega_m, i\nu_o, i\nu_{o'}, \mathbf{q}) = D_{n,n'}(i\omega_m, -i\nu_o, -i\nu_{o'}, -\mathbf{q}) \tag{63b}$$

$$C^*_{n,n'}(i\omega_m, i\nu_o, i\nu_{o'}, \mathbf{q}) = C_{n',n}(-i\omega_m, -i\nu_{o'}, -i\nu_o, \mathbf{q}) \, . \tag{63c}$$

### A.3  Connection between $\mathcal{K}_2$ and $\bar{\mathcal{K}}_2$

In Section 3.1.2 we argued that $\bar{\mathcal{K}}_2$ can be obtained from $\mathcal{K}_2$ by symmetry. For the $pp$ and $\overline{ph}$ channel the time reversal symmetry exchanges the two fermionic dependencies while keeping the transfer frequency and momentum fixed. The same holds for the $ph$-channel by using the combination of the crossing and the time reversal symmetry. Taking the limit of large frequencies for the first and second fermionic frequency respectively, we obtain trivially

$$\mathcal{K}_{2,P,n}(i\omega_m, i\nu_o, \mathbf{q}) = \bar{\mathcal{K}}_{2,P,n}(i\omega_n, i\nu_o, \mathbf{q}) \tag{64a}$$

$$\mathcal{K}_{2,D,n}(i\omega_m, i\nu_o, \mathbf{q}) = \bar{\mathcal{K}}_{2,D,n}(i\omega_n, i\nu_o, \mathbf{q}) \tag{64b}$$

$$\mathcal{K}_{2,C,n}(i\omega_m, i\nu_o, \mathbf{q}) = \bar{\mathcal{K}}_{2,C,n}(i\omega_n, i\nu_o, \mathbf{q}) \, . \tag{64c}$$

## B  Formal derivation of the fRG flow equations for $\chi$ and $\gamma_3$

In this section we provide an explicit derivation of the flow equations for the response functions. As anticipated in Sec. 2.2, we start by coupling the fermionic bilinears to an external source field $J$, by adding the following scalar product

$$(J^n_\eta, \rho^n_\eta) = \int dk J^n_\eta(k) \rho^n_\eta(k) \, , \tag{65}$$

where $n$ indicates the momentum structure of the fermionic bilinears coupled to the field $J^n_\eta$. Since the density is in general not charge conserving, it is convenient to use the Nambu formalism that allows for a more concise derivation of the flow equations of the physical response functions. We rewrite Eqs. (1) and (2) in the Nambu basis [94, 95]

$$\rho^n_\eta(q) = \sum_{s,s'=\pm} \alpha^\eta_{s,s'} \int dp \, \bar{\phi}_s(p-q) f_n(\mathbf{p}) \phi_{s'}(p) \, , \tag{66}$$

where $s = \pm$ represents the Nambu index and

$$\phi_+(k) = \psi_\uparrow(k) \qquad\qquad \bar{\phi}_+(k) = \bar{\psi}_\uparrow(k)$$

$$\phi_-(k) = \bar{\psi}_\downarrow(-k) \qquad\qquad \bar{\phi}_-(k) = \psi_\downarrow(-k) \, .$$

The matrices $\boldsymbol{\alpha}^\eta$ (with $\eta = \{\mathrm{d}, \mathrm{m}, \mathrm{sc}\}$), which define the Nambu index structure in the different physical channels, are given by

$$\boldsymbol{\alpha}^{\mathrm{d}} = \begin{pmatrix} 1 & 0 \\ 0 & -1 \end{pmatrix} \quad \boldsymbol{\alpha}^{\mathrm{m}} = \begin{pmatrix} 1 & 0 \\ 0 & 1 \end{pmatrix} \quad \boldsymbol{\alpha}^{\mathrm{sc}} = \begin{pmatrix} 0 & 1 \\ -1 & 0 \end{pmatrix}. \tag{67}$$

In order to derive the flow equations for the fermion-boson vertex of Eq. (15) and the susceptibility of Eq. (14) we start from the so-called Wetterich equation [96]

$$\partial_\Lambda \Gamma^\Lambda[J_\eta, \phi] = -(\bar{\phi}, \dot{Q}_0^\Lambda \phi) - \frac{1}{2} \mathrm{tr}\left\{ \dot{\mathbf{Q}}_0^\Lambda (\boldsymbol{\Gamma}^{(2)\Lambda}[J_\eta, \phi])^{-1} \right\}, \tag{68}$$

where $\Gamma^\Lambda$ represents the scale-dependent effective action, which is a function of the functional variable $J_\eta$ and the Nambu field $\phi$, $Q_0^\Lambda$ is the inverse non-interacting Green's function and the dot denotes the derivative with respect to the flow parameter $\Lambda$. Further, the matrix $\mathbf{Q}_0^\Lambda = \mathrm{diag}(Q_0^\Lambda, -Q_0^{\Lambda,t})$ and

$$\boldsymbol{\Gamma}^{(2)\Lambda}[J_\eta, \phi] = \begin{pmatrix} \bar{\partial}\partial\Gamma^\Lambda[J_\eta, \phi] & \bar{\partial}\bar{\partial}\Gamma^\Lambda[J_\eta, \phi] \\ \partial\partial\Gamma^\Lambda[J_\eta, \phi] & \partial\bar{\partial}\Gamma^\Lambda[J_\eta, \phi] \end{pmatrix} \tag{69}$$

were we used, where $\partial$ and $\bar{\partial}$ applied to the effective action $\Gamma^\Lambda$ are a shorthand notation for the functional derivative with respect to $\phi$ and $\bar{\phi}$, respectively. Following the derivation of Ref. [2], we introduce the matrix

$$\mathbf{U}^\Lambda[J_\eta, \phi] = (\mathbf{G}^\Lambda)^{-1} - \boldsymbol{\Gamma}^{(2)\Lambda}[J_\eta, \phi]. \tag{70}$$

Thus, we can recast $(\boldsymbol{\Gamma}^{(2)\Lambda}[J_\eta, \phi])^{-1} = (\mathbf{1} - \mathbf{G}^\Lambda \mathbf{U}^\Lambda)^{-1} \mathbf{G}^\Lambda$ and expand the inverse matrix in a geometric series

$$(\boldsymbol{\Gamma}^{(2)\Lambda}[J_\eta, \phi])^{-1} = \sum_{n=0}^{\infty} (\mathbf{G}^\Lambda \mathbf{U}^\Lambda)^n \, \mathbf{G}^\Lambda. \tag{71}$$

We can now insert Eq. (71) in Eq. (68). Expanding up to second order yields

$$\partial_\Lambda \Gamma^\Lambda[J_\eta, \phi] = -(\bar{\phi}, \dot{Q}_0^\Lambda \phi) - \frac{1}{2} \mathrm{tr}\left\{ \dot{\mathbf{Q}}_0^\Lambda \mathbf{G}^\Lambda \right\} - \frac{1}{2} \mathrm{tr}\left\{ \mathbf{S}^\Lambda \mathbf{U}^\Lambda \right\} - \frac{1}{2} \mathrm{tr}\left\{ \mathbf{S}^\Lambda \mathbf{U}^\Lambda \mathbf{G}^\Lambda \mathbf{U}^\Lambda \right\} + ..., \tag{72}$$

where $\mathbf{S}^\Lambda = \mathbf{G}^\Lambda \mathbf{Q}_0^\Lambda \mathbf{G}^\Lambda = \mathrm{diag}(S^\Lambda, -S^{\Lambda,t})$ represents the matrix diagonal form of the single scale propagator, and we exploited the cyclic property of the trace. After applying the trace to the matrices in the curly brackets, we can expand the effective action in powers of the fermionic Nambu fields and the external bosonic source field

$$\Gamma^\Lambda[J_\eta, \phi] = \sum_{m_1, n_1 = 0}^{\infty} \frac{(-1)^{m_1}}{n_1! \, (m_1!)^2} \times$$

$$\sum_{\substack{x_1...x_{m_1} \\ x_1'...x_{m_1}' \\ y_1...y_{n_1}}} \frac{\partial^{(2m_1 + n_1)} \Gamma^\Lambda[J_\eta, \phi]}{\partial J_\eta(y_1)...\partial J_\eta(y_{n_1}) \partial \bar{\phi}(x_1')...\partial \bar{\phi}(x_{m_1}') \partial \phi(x_{m_1})...\phi(x_1)} \Big|_{\phi = J_\eta = 0} \times$$

$$J_\eta(y_1)...J_\eta(y_{n_1}) \bar{\phi}(x_1')...\bar{\phi}(x_{m_1}') \phi(x_{m_1})...\phi(x_1) \tag{73a}$$

$$= \sum_{m_1, n_1 = 0}^{\infty} \frac{(-1)^{m_1}}{n_1! \, (m_1!)^2} \sum_{\substack{x_1...x_{m_1} \\ x_1'...x_{m_1}' \\ y_1...y_{n_1}}} \gamma^\Lambda_{2m_1 + n_1, y_1..y_{n_1}, x_1'..x_{m_1}', x_1...x_{m_1}} \times$$

$$J_\eta(y_1)...J_\eta(y_{n_1}) \bar{\phi}(x_1')...\bar{\phi}(x_{m_1}') \phi(x_{m_1})...\phi(x_1). \tag{73b}$$

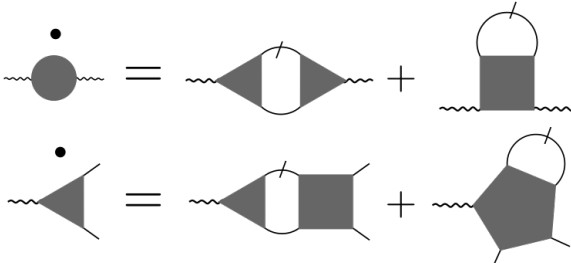

Figure 15: Simplified diagrammatic representation of the flow equations for the susceptibily (first line) and the fermion-boson vertex (second line) illustrating the topological structure of the diagrams. The circle, triangle and the square represent the susceptibility $\chi$, the fermion-boson vertex $\gamma_3$, and the two-particle vertex $\gamma_4$, respectively.

Note that the index $x = \{s, k\}$ combines the Nambu index $s$ and the fermionic quadrivector $k = (\nu, \mathbf{k})$ (here we disregard additional quantum numbers, as e.g., orbital), while $y = \{n, q\}$ combines the momentum structure of the coupling to the bilinears, $n$, with the bosonic quadrivector $q = (\omega, \mathbf{q})$. Inserting this expansion in Eq. (72), we compare the expansion coefficient related to the same order on the fields on both sides of the equation.

For $n_1 = 0$ we recover the standard fermionic hierarchy of flow equations [1,2]. For $n_1 > 0$ we can derive the flow equations for the fermion-boson vertex ($n_1 = 1$, $m_1 = 1$) as well as for the boson-boson vertices or susceptibilities ($n_1 = 2$, $m_1 = 0$). In Nambu notation, the flow equation for the susceptibility reads

$$\partial_\Lambda \chi^\Lambda(y, y') = \sum_{\substack{x_1, x_1' \\ x_2, x_2'}} \gamma_3^\Lambda(y, x_1', x_1)[G^\Lambda(x_1, x_2')S^\Lambda(x_2, x_1') + (S \leftrightarrow G)]\gamma_3^{\Lambda\dagger}(y', x_2, x_2') +$$
$$\sum_{x_1, x_1'} S^\Lambda(x_1, x_1')\tilde{\gamma}_4^\Lambda(y, y', x_1', x_1), \tag{74}$$

and the one for the fermion-boson vertex is

$$\partial_\Lambda \gamma_3^\Lambda(y, x', x) = \sum_{x_1, x_1'} S^\Lambda(x_1, x_1')\gamma_5^\Lambda(y, x', x_1', x_1, x) +$$
$$\sum_{\substack{x_1, x_1' \\ x_2, x_2'}} \gamma_3^\Lambda(y, x_1', x_1)[G^\Lambda(x_1, x_2')S^\Lambda(x_2, x_1') + (S \leftrightarrow G)]\gamma_4^\Lambda(x', x_2, x_2', x). \tag{75}$$

In the second term on the r.h.s. of Eq. (74), $\gamma_{2m_1+n_1}^\Lambda = \tilde{\gamma}_{2+2}^\Lambda$ represents the functional derivative of the effective action with respect to two bosonic and two fermionic Nambu fields. The two Eqs. (74) and (75) are schematically shown in Fig. 15. If one neglects the second term in both r.h.s., they correspond to the $1\ell$ fRG equations for the response functions. Both $\gamma_3$ and $\chi$ do not feed back into the flow equations for $\gamma_4$ and $\Sigma$.

## C Connection between the vertex asymptotics and the response functions

In this appendix we demonstrate that the integration of the fRG flow equations for the so-called kernel functions $\mathcal{K}_1$ and $\mathcal{K}_2$ mentioned in Section 3, coincide with the $s$-wave susceptibility

and fermion-boson vertex resulting from the flow.

Let us write explicitly the flow equation for the asymptotics $\mathcal{K}_{1,\eta}^{\Lambda}$ and $\bar{\mathcal{K}}_{2,\eta}^{\Lambda}$, with $\eta = \{\text{sc,d,m}\}$, obtained from Eq. (43) in the limit of infinite fermionic Matsubara frequencies $\nu$ and $\nu'$

$$\lim_{\substack{\nu\to\infty \\ \nu'\to\infty}} \dot{\boldsymbol{\phi}}_{\eta}^{\Lambda} = \dot{\mathcal{K}}_{1,\eta}^{\Lambda} = (\boldsymbol{\gamma}_{4,\eta}^{0} + \mathcal{K}_{1,\eta}^{\Lambda} + \bar{\mathcal{K}}_{2,\eta}^{\Lambda}) \circ \dot{\boldsymbol{\Pi}}_{\eta}^{\Lambda} \circ (\boldsymbol{\gamma}_{4,\eta}^{0} + \mathcal{K}_{1,\eta}^{\Lambda} + \mathcal{K}_{2,\eta}^{\Lambda}) +$$

$$(\boldsymbol{\gamma}_{4,\eta}^{0} + \mathcal{K}_{1,\eta}^{\Lambda} + \bar{\mathcal{K}}_{2,\eta}^{\Lambda}) \circ \boldsymbol{\Pi}_{\eta}^{\Lambda} \circ \dot{\mathbf{i}}_{\eta}^{\Lambda} \circ \boldsymbol{\Pi}_{\eta}^{\Lambda} \circ (\boldsymbol{\gamma}_{4,\eta}^{0} + \mathcal{K}_{1,\eta}^{\Lambda} + \mathcal{K}_{2,\eta}^{\Lambda}) \qquad (76)$$

and

$$\lim_{\nu\to\infty} \dot{\boldsymbol{\phi}}_{\eta}^{\Lambda} = \dot{\mathcal{K}}_{1,\eta}^{\Lambda} + \dot{\bar{\mathcal{K}}}_{2,\eta}^{\Lambda} = (\boldsymbol{\gamma}_{4,\eta}^{0} + \mathcal{K}_{1,\eta}^{\Lambda} + \bar{\mathcal{K}}_{2,\eta}^{\Lambda}) \circ \dot{\boldsymbol{\Pi}}_{\eta}^{\Lambda} \boldsymbol{\gamma}_{4,\eta}^{\Lambda} +$$

$$(\boldsymbol{\gamma}_{4,\eta}^{0} + \mathcal{K}_{1,\eta}^{\Lambda} + \bar{\mathcal{K}}_{2,\eta}^{\Lambda}) \circ \boldsymbol{\Pi}_{\eta}^{\Lambda} \circ \dot{\mathbf{i}}_{\eta}^{\Lambda} +$$

$$(\boldsymbol{\gamma}_{4,\eta}^{0} + \mathcal{K}_{1,\eta}^{\Lambda} + \bar{\mathcal{K}}_{2,\eta}^{\Lambda}) \circ \boldsymbol{\Pi}_{\eta}^{\Lambda} \circ \dot{\mathbf{i}}_{\eta}^{\Lambda} \circ \boldsymbol{\Pi}_{\eta}^{\Lambda} \circ \boldsymbol{\gamma}_{4,\eta}^{\Lambda}, \qquad (77)$$

where $\dot{\boldsymbol{\phi}}_{\eta}^{\Lambda}$ is given by

$$\dot{\boldsymbol{\phi}}_{\text{sc}}^{\Lambda} = \dot{\boldsymbol{P}}^{\Lambda} \qquad (78a)$$

$$\dot{\boldsymbol{\phi}}_{\text{d}}^{\Lambda} = 2\dot{\boldsymbol{D}}^{\Lambda} - \dot{\boldsymbol{C}}^{\Lambda} \qquad (78b)$$

$$\dot{\boldsymbol{\phi}}_{\text{m}}^{\Lambda} = -\dot{\boldsymbol{C}}^{\Lambda}, \qquad (78c)$$

the bare vertex $\boldsymbol{\gamma}_{4,\eta}^{0} = \mp U$ corresponds to the Hubbard interaction (with the minus sign for $\eta = \text{sc, d}$ and the plus sign for $\eta = \text{m}$), and the asymptotic vertex function $\bar{\mathcal{K}}_{2,\eta}^{\Lambda}$ is related to $\mathcal{K}_{2,\eta}^{\Lambda}$ by symmetry (see Appendix A). For local bare interactions, the only non-zero elements of the matrices $\dot{\mathcal{K}}_{1,\eta}^{\Lambda}$ and $\boldsymbol{\gamma}_{4,\eta}^{0}$ correspond to both form factors being equal to zero, and of $\mathcal{K}_{2,\eta}^{\Lambda}$ ($\bar{\mathcal{K}}_{2,\eta}^{\Lambda}$) to a vanishing second (first) form factor.

The connection between the vertex asymptotics and the response function is shown by induction using the assumption

$$\boldsymbol{\gamma}_{4,\eta}^{0} + \mathcal{K}_{1,\eta}^{\Lambda} + \bar{\mathcal{K}}_{2,\eta}^{\Lambda} = \alpha \boldsymbol{\gamma}_{3,\eta}^{\Lambda}(\omega, \nu, \mathbf{q}). \qquad (79)$$

For the initial condition, it holds $\boldsymbol{\gamma}_{3,\eta}^{\Lambda_{\text{init}}} = \boldsymbol{\gamma}_{3,\eta}^{0} = \mathbb{1}$. Since $\mathcal{K}_{1}^{\Lambda_{\text{init}}}$ and $\mathcal{K}_{2}^{\Lambda_{\text{init}}}$ both vanish, one has $\alpha = \boldsymbol{\gamma}_{4,\eta}^{0} = \mp U \delta_{n,0} \delta_{n',0}$. Considering $(\boldsymbol{\gamma}_{4,\eta}^{0} + \mathcal{K}_{1,\eta}^{\Lambda} + \bar{\mathcal{K}}_{2,\eta}^{\Lambda})$ for an arbitrary value of $\Lambda$, we can identify the flow equation of the asymptotics with the one of $\gamma_3$, see Eq. (23a). Therefore Eq. (79) applies also for the following $\Lambda$ step. As a consequence we can extract the fermion-boson vertex from the vertex asymptotics. Finally, inserting Eq. (79) into (76), we obtain the flow equation for the susceptibility (23a).

The $s$-wave fRG results for the susceptibility and the fermion-boson vertex can be extracted from the asymptotic vertex functions $\mathcal{K}_{1,\eta}^{\Lambda}$ and $\bar{\mathcal{K}}_{2,\eta}^{\Lambda}$ by dividing the $s$-wave form factor component by the bare interaction $\mp U$. Since $\boldsymbol{\gamma}_{4,\eta}^{0}$ vanishes for all other form factor combinations, other than $s$-wave response functions cannot be recovered by the asymptotics. This observation simplifies the fRG implementation, where the flow equations for $\chi$ and $\gamma_3$ can be omitted if only their $s$-wave components are needed.

## D  "Post-processed" flow equations for $\gamma_3$ and $\chi$

In this section we explicitly provide the scale derivative of Eqs. (8) and (10) for the case in which the $\Sigma$ and $\gamma_4$ entering the r.h.s. are obtained from the integration of the corresponding

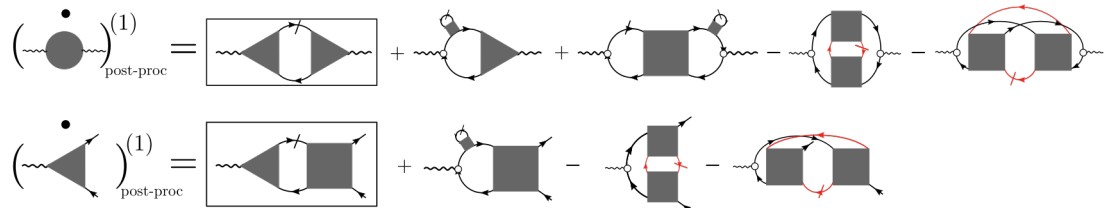

Figure 16: Diagrammatic representation of Eqs. (81) (first line) and (83) (second line), where the boxes indicate the conventional $1\ell$ approximation. The internal loops in red provide the particle-hole and particle-particle contributions respectively. The empty dot represents the bare AF fermion-boson vertex $(\gamma^0_{3,m})_{n,m} = \delta_{n,m}$.

$1\ell$ flow equations. We first consider Eq. (10) and, after introducing a $\Lambda$-dependence of the Green's functions and of $\gamma_4$ on the r.h.s., perform the full derivative with respect to $\Lambda$. For simplicity we here consider the magnetic vertex as example, which is directly related to the particle-hole crossed vertex by

$$\gamma_{4,\mathrm{m}}(q,k,k') = \gamma_{4,ph,\uparrow\uparrow}(q,k,k') - \gamma_{4,ph,\uparrow\downarrow}(q,k,k') = \gamma_{4,ph,\bar{\uparrow}\downarrow}(k'-k,k,k+q) = -\gamma_{4,\overline{ph},\uparrow\downarrow}(q,k,k'),$$
(80)

where we used the SU(2) and crossing symmetries [32]. The derivative of the fermion-boson vertex with respect to $\Lambda$, as obtained from Eq. (10), reads

$$\begin{aligned}
\partial_\Lambda \left( \gamma^\Lambda_{3,\mathrm{m}} \right)^{(1)}_{\text{post-proc}} &= \partial_\Lambda \left( \gamma^0_{3,\mathrm{m}} + \gamma^0_{3,\mathrm{m}} \circ \mathbf{\Pi}^\Lambda_\mathrm{m} \circ \gamma^\Lambda_{4,\mathrm{m}} \right) \\
&= \gamma^0_{3,\mathrm{m}} \circ \left( \dot{\mathbf{\Pi}}^{\Lambda,(1)}_{S,\mathrm{m}} + \dot{\tilde{\mathbf{\Pi}}}^{\Lambda,(1)}_\mathrm{m} \right) \circ \gamma^\Lambda_{4,\mathrm{m}} + \gamma^0_{3,\mathrm{m}} \circ \mathbf{\Pi}^\Lambda_\mathrm{m} \circ \dot{\gamma}^{\Lambda,(1)}_{4,\mathrm{m}} \\
&= \gamma^0_{3,\mathrm{m}} \circ \left( \dot{\mathbf{\Pi}}^{\Lambda,(1)}_{S,\mathrm{m}} + \dot{\tilde{\mathbf{\Pi}}}^{\Lambda,(1)}_\mathrm{m} \right) \circ \gamma^\Lambda_{4,\mathrm{m}} - \\
&\quad \gamma^0_{3,\mathrm{m}} \circ \mathbf{\Pi}^\Lambda_\mathrm{m} \circ \left( \dot{\mathbf{C}}^{\Lambda,(1)} - \hat{\mathbf{C}}[\dot{\phi}^{\Lambda,(1)}_{ph}] - \hat{\mathbf{C}}[\dot{\phi}^{\Lambda,(1)}_{pp}] \right) \\
&= \gamma^\Lambda_{3,\mathrm{m}} \circ \dot{\mathbf{\Pi}}^{\Lambda,(1)}_{S,\mathrm{m}} \circ \gamma^\Lambda_{4,\mathrm{m}} + \gamma^0_{3,\mathrm{m}} \circ \dot{\tilde{\mathbf{\Pi}}}^{\Lambda,(1)}_\mathrm{m} \circ \gamma^\Lambda_{4,\mathrm{m}} - \gamma^0_{3,\mathrm{m}} \circ \mathbf{\Pi}^\Lambda_\mathrm{m} \circ \hat{\mathbf{C}}[\dot{\phi}^{\Lambda,(1)}_{ph}] - \\
&\quad \gamma^0_{3,\mathrm{m}} \circ \mathbf{\Pi}^\Lambda_\mathrm{m} \circ \hat{\mathbf{C}}[\dot{\phi}^{\Lambda,(1)}_{pp}],
\end{aligned}$$
(81)

where for sake of conciseness we used a tensor-product form. In contrary to the definition in Sec. 3.2, the bubble $\mathbf{\Pi}^\Lambda_{S,\mathrm{m}}$ does not have the Katanin substitution [45] and we define $\dot{\tilde{\mathbf{\Pi}}}_\mathrm{m} = \dot{\mathbf{\Pi}}_{S \to G\dot{\Sigma}G,\mathrm{m}}$ in order to take care of the scale derivative in the self-energy. Further $\gamma^0_{3,\mathrm{m}} = \mathbb{1}$, and $\hat{\mathbf{C}}[\dot{\phi}_\eta]$ stands for

$$\hat{\mathbf{C}}[\dot{\phi}^\Lambda_{ph}]_{n,n'} = \int d\mathbf{k}d\mathbf{k}' f^*_n(\mathbf{k}) f_{n'}(\mathbf{k}') \dot{\phi}^\Lambda_{ph}(k'-k,k,k+q) \tag{82a}$$

$$\hat{\mathbf{C}}[\dot{\phi}^\Lambda_{pp}]_{n,n'} = \int d\mathbf{k}d\mathbf{k}' f^*_n(\mathbf{k}) f_{n'}(\mathbf{k}') \dot{\phi}^\Lambda_{pp}(q+k+k',k,k+q). \tag{82b}$$

The superscript (1) indicates that flowing objects ($\Sigma$ and the $\phi$'s) are computed within $1\ell$ from their corresponding differential equations. From the second to the third line of Eq. (81) we used Eq. (80) and the parquet decomposition in Eq. (32). The diagrammatic representation of the last line of Eq. (81) is shown in the first line of Fig. 16.

Let us now turn to Eq. (8) for the susceptibility, where we again restrict ourselves to the

magnetic channel. Following the derivation of Eq. (81) one obtains

$$
\begin{aligned}
\partial_\Lambda \left( \boldsymbol{\chi}_{\mathrm{m}}^\Lambda \right)_{\text{post-proc}}^{(1)} &= \partial_\Lambda \left( \boldsymbol{\gamma}_{3,\mathrm{m}}^0 \circ \boldsymbol{\Pi}_{\mathrm{m}}^\Lambda \circ \boldsymbol{\gamma}_{3,\mathrm{m}}^{\dagger,0} + \boldsymbol{\gamma}_{3,\mathrm{m}}^0 \circ \boldsymbol{\Pi}_{\mathrm{m}}^\Lambda \circ \boldsymbol{\gamma}_{4,\mathrm{m}}^\Lambda \circ \boldsymbol{\Pi}_{\mathrm{m}}^\Lambda \circ \boldsymbol{\gamma}_{3,\mathrm{m}}^{\dagger,0} \right) \\
&= \boldsymbol{\gamma}_{3,\mathrm{m}}^\Lambda \circ \dot{\boldsymbol{\Pi}}_{S,\mathrm{m}}^{\Lambda,(1)} \circ \boldsymbol{\gamma}_{3,\mathrm{m}}^{\Lambda,\dagger} + \\
&\quad \boldsymbol{\gamma}_{3,\mathrm{m}}^0 \circ \dot{\bar{\boldsymbol{\Pi}}}_{\mathrm{m}}^{\Lambda,(1)} \circ \boldsymbol{\gamma}_{3,\mathrm{m}}^{\Lambda,\dagger} + \boldsymbol{\gamma}_{3,\mathrm{m}}^0 \circ \boldsymbol{\Pi}_{\mathrm{m}}^\Lambda \circ \boldsymbol{\gamma}_{4,\mathrm{m}}^\Lambda \circ \dot{\bar{\boldsymbol{\Pi}}}_{\mathrm{m}}^{\Lambda,(1)} \circ \boldsymbol{\gamma}_{3,\mathrm{m}}^{\dagger,0} - \\
&\quad \boldsymbol{\gamma}_{3,\mathrm{m}}^0 \circ \boldsymbol{\Pi}_{\mathrm{m}}^\Lambda \circ \hat{\boldsymbol{C}}[\dot{\boldsymbol{\phi}}_{ph}^{\Lambda,(1)}] \circ \boldsymbol{\Pi}_{\mathrm{m}}^\Lambda \circ \boldsymbol{\gamma}_{3,\mathrm{m}}^{\dagger,0} - \\
&\quad \boldsymbol{\gamma}_{3,\mathrm{m}}^0 \circ \boldsymbol{\Pi}_{\mathrm{m}}^\Lambda \circ \hat{\boldsymbol{C}}[\dot{\boldsymbol{\phi}}_{pp}^{\Lambda,(1)}] \circ \boldsymbol{\Pi}_{\mathrm{m}}^\Lambda \circ \boldsymbol{\gamma}_{3,\mathrm{m}}^{\dagger,0} ,
\end{aligned}
\tag{83}
$$

with the diagrammatic representation is provided in Fig. 16 (second line). One observes the appearance of additional terms on the r.h.s. of the post-processing flow equations for $\gamma_3$ and $\chi$ with respect to their standard $1\ell$ equations, indicated by the boxes in Fig. 16. Besides the terms containing the $\Lambda$ derivative of the self-energy (which are included in the Katanin corrections [45]), let us draw the attention to the last two diagrams appearing on the r.h.s. for both $\partial_\Lambda(\gamma_3^\Lambda)_{\text{post-proc}}$ and $\partial_\Lambda(\chi^\Lambda)_{\text{post-proc}}$. The diagrammatic structure in terms of loops is of second order for $\gamma_3$ and of third for $\chi$. The integration of these post-processed flow equations, along with the $1\ell$ flow equations for $\Sigma$ and $\gamma_4$, would generate the last two diagrams already at the first integration step $\Lambda_{\text{init}} + d\Lambda$ (with $d\Lambda < 0$ in the $\Omega$-flow), providing the following contribution to $\dot{\boldsymbol{\chi}}_{\mathrm{m}}^{\Lambda_{\text{init}}}$

$$
-\boldsymbol{\gamma}_{3,\mathrm{m}}^0 \circ \left( \boldsymbol{\Pi}_{\mathrm{m}}^{\Lambda_{\text{init}}} \circ \hat{\boldsymbol{C}}[\dot{\boldsymbol{\phi}}_{ph}^{\Lambda_{\text{init}}}] \circ \boldsymbol{\Pi}_{\mathrm{m}}^{\Lambda_{\text{init}}} + \boldsymbol{\Pi}_{\mathrm{m}}^{\Lambda_{\text{init}}} \circ \hat{\boldsymbol{C}}[\dot{\boldsymbol{\phi}}_{pp}^{\Lambda_{\text{init}}}] \circ \boldsymbol{\Pi}_{\mathrm{m}}^{\Lambda_{\text{init}}} \right) \circ \boldsymbol{\gamma}_{3,\mathrm{m}}^{\dagger,0} .
\tag{84}
$$

The first term vanishes due to the Pauli principle ($\dot{\boldsymbol{\phi}}_{ph}^{\Lambda_{\text{init}}} = 0$, see Ref. [34]), and the last one provides a negative contribution which reduces the $1\ell$ term. In fact, the unscreened particle-particle bubble entering $\hat{C}[\dot{\phi}_{pp}^{\Lambda_{\text{init}}}]_{n,m}$ has the same sign of the unscreened (magnetic) $S - G$ bubble. This overall suppression by the additional $3\ell$-like terms is a general feature of the post-processed fRG scheme. The unbalance between the $1\ell$ $\gamma_4$ flow, which topologically cuts part of the parquet diagrams, and the additional $3\ell$-like diagrams of the susceptibility flow, leads to an artificial overscreening of the conventional $1\ell$ calculation. Analogous conclusions can be drawn for the density and superconducting channels. Thus one expects a pronounced effect in the secondary channels because the dominant channel enters the internal loop of one of the two $3\ell$-like additional diagrams, resulting in a reduction with respect to the converged data. In contrast, the dominant channel will not be affected that strongly, presenting only a slight overestimation of the post-processed susceptibility at the $1\ell$ level (see Fig. 8). Moreover, since this overscreening affects all frequencies, it may be responsible for the unphysical negative value of the density susceptibility observed at finite frequencies in Fig. 9. In particular, since the parquet diagrams disregarded in the $1\ell$ approximation depend on the cutoff, the detected unphysical results in the secondary channels were observed to be more severe for the interaction flow. We finally note that this opposite effect of the density and the superconducting channels with respect to the dominant magnetic channel has been observed also in Ref. [97] by analyzing the effect of the parquet decomposition of the vertex on the self-energy.

## E  Two-loop approximation for $\gamma_3$'s flow equation

We here provide the derivation of the $2\ell$ corrections to the conventional $1\ell$ truncated flow equations. The derivation follows the scheme adopted for the flow equation of the two-particle vertex as reported in Ref. [17]. Our goal is to include the feedback of $\gamma_5^\Lambda$ onto the flow equation for $\gamma_3^\Lambda$, see Eq. (75), at the second order in the effective interaction. From the derivation

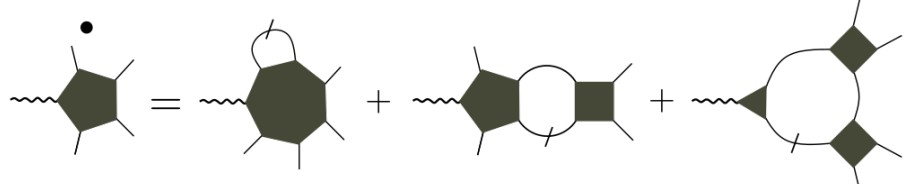

Figure 17: Simplified diagrammatic representation of the flow equation for $\gamma_5^\Lambda$ illustrating the topological structure of the diagrams.

provided in Appendix B, one sees that the differential equation for $\gamma_5^\Lambda$ is given by the sum of all diagrams which have the topological structure depicted in Fig. 17. The first and the second diagrams on the r.h.s. are at least of third order in the effective interaction since $\gamma_7^\Lambda$ (depicted by a heptagon) and $\gamma_5$ (depicted by a pentagon) are at least $O((\gamma_4^\Lambda)^3)$ and $O((\gamma_4^\Lambda)^2)$, respectively. Therefore, we can restrict ourselves to diagrams with a topological structure of the third one. Its contribution can be obtained by taking the following functional derivative evaluated at zero fields

$$\partial_\Lambda \gamma_5^\Lambda(y, x_1', x_2, x_1, x_2) = \frac{\partial^5}{\partial J_\eta(y) \partial \bar{\phi}(x_1') \partial \bar{\phi}(x_2') \partial \phi(x_2) \partial \phi(x_1)} \times$$

$$\left[ \frac{1}{3} \partial_{\Lambda,S} \mathrm{tr}\left(G^\Lambda \bar{\partial} \partial \Gamma^\Lambda G^\Lambda \bar{\partial} \partial \Gamma^\Lambda G^\Lambda \bar{\partial} \partial \Gamma^\Lambda\right) - \partial_{\Lambda,S} \mathrm{tr}\left(G^\Lambda \bar{\partial} \partial \Gamma^\Lambda G^\Lambda \bar{\partial} \bar{\partial} \Gamma^\Lambda G^{\Lambda,t} \partial \partial \Gamma^\Lambda\right) \right]\Big|_{J=\phi=0}, \tag{85}$$

where $x = \{s, k\}$, $y = \{\eta, q\}$ and $\partial_{\Lambda,S}$ acts only on $G^\Lambda$ and returns the single-scale propagator $S^\Lambda$. At this point we integrate the r.h.s. which is an easy operation once we take into account that i) one can replace $S^\Lambda = \partial_{\Lambda,S} G^\Lambda$ by the full derivative $\partial_\Lambda G^\Lambda$ since their difference due the derivative of the self-energy is of higher order in the effective interaction $\gamma_4^\Lambda$, and ii) one can let the scale derivative act also on $\gamma_4^\Lambda$ since its derivative is at least of order $O((\gamma_4^\Lambda)^2)$. According to these arguments, the r.h.s. of Eq. (85) can be approximated by the total derivative with respect to the $\Lambda$ and integrated to

$$\gamma_5^\Lambda(y, x_1', x_2, x_1, x_2) = \frac{\partial^5}{\partial J_\eta(y) \partial \bar{\phi}(x_1') \partial \bar{\phi}(x_2') \partial \phi(x_2) \partial \phi(x_1)}$$

$$\left[ \frac{1}{3} \mathrm{tr}\left(G^\Lambda \bar{\partial} \partial \Gamma^\Lambda G^\Lambda \bar{\partial} \partial \Gamma^\Lambda G^\Lambda \bar{\partial} \partial \Gamma^\Lambda\right) - \mathrm{tr}\left(G^\Lambda \bar{\partial} \partial \Gamma^\Lambda G^\Lambda \bar{\partial} \bar{\partial} \Gamma^\Lambda G^{\Lambda,t} \partial \partial \Gamma^\Lambda\right) \right]\Big|_{J=\phi=0}. \tag{86}$$

The only terms surviving the functional derivative are all connected diagrams composed by two two-particle vertices $\gamma_4^\Lambda$ and one fermion-boson vertex $\gamma_3^\Lambda$. What distinguishes the first and the second contributions of Eq. (86) is the position of $\gamma_3^\Lambda$ which can be inserted at all $\bar{\partial} \partial$ in the first line, while is restricted to a single $\bar{\partial} \partial$ in the second one because of the conservation of Nambu particles. Moreover, the first term accounts for two-particle vertices whose external lines are always a particle and a hole, whereas in the second term they are attached to two particles $\partial \partial \Gamma^\Lambda$ and two holes $\bar{\partial} \bar{\partial}$, respectively. The topological structure of these two contributions is schematically shown in Fig. 18.

The last step consists in closing these diagrams in all possible ways by means of the single-scale propagator and adding them to the flow equation of $\gamma_3^\Lambda$. Hence, one obtains $2\ell$ approximated flow equations for $\gamma_3^\Lambda$ which contain terms of the order $O((\gamma_4^\Lambda)^2)$ in the effective interaction. We can classify [17, 40] the $2\ell$ corrections according to their topological structure, with overlapping loops (Fig. 19 (b)) and non-overlapping loops (Fig. 19 (a)). We observe that the latter can be included in the $1\ell$ equations by using the Katanin correction [45] where

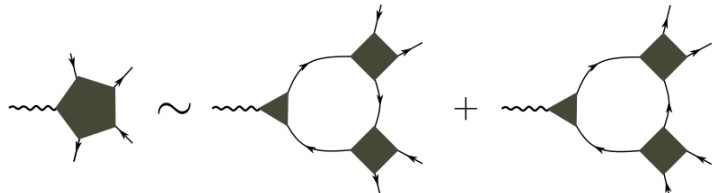

Figure 18: Diagrammatic contributions for $\gamma_5^\Lambda$ up to the second oder in the effective interaction $\gamma_4^\Lambda$.

$S^\Lambda \to S^\Lambda + G^\Lambda \dot{\Sigma}^\Lambda G^\Lambda$. The remaining $2\ell$ corrections have as building block the $1\ell$ diagrams of the flow equation of $\gamma_4^\Lambda$. Translating our Nambu formalism to the physical fields, those corrections yield Eq. (24).

## F  Implementation details

Here we provide the explicit form of $\boldsymbol{\gamma}_{4,\{P,D,C\}}$ appearing on the r.h.s. of Eq. (36). By using the parquet decomposition in the diagrammatic channels (see Eq. (32)), the first contribution of the projections of the four-point vertex onto the different channels is the projection of the fully two-particle irreducible vertex, approximated by its first order in the on-site Hubbard interaction $U$, onto the form-factor basis. The projected bare interaction is

$$[\hat{P}[U](i\omega_l, i\nu_o, i\nu_{o'}, \mathbf{q})]_{n,n'} = [\hat{D}[U](i\omega_l, i\nu_o, i\nu_{o'}, \mathbf{q})]_{n,n'}$$
$$= [\hat{C}[U](i\omega_l, i\nu_o, i\nu_{o'}, \mathbf{q})]_{n,n'} = -U\delta_{n,0}\delta_{n',0} . \qquad (87)$$

Secondly, every channel, written in its natural bosonic-fermionic notation on the l.h.s. of Eq. (36), needs to be projected onto the complementary channels. The projection of one channel $\phi_r$ to another leads to a linear combination of its frequency arguments (see Eq. (22) for the physical channels and Eq. (91a) to (91f) for the diagrammatic channels). In momentum space, the projection is more involved due to the form factor dependence. Following the procedure of Ref. [31], we identify the projection matrices which describe the momentum translation from one channel to another using a matrix multiplication

$$[\hat{B}[\phi_{B'}](i\omega_l, i\nu_o, i\nu_{o'}, \mathbf{q})]_{n,n'} = \sum_{m,m',\mathbf{l}} A_{n,n',m,m'}^{B,B'}(\mathbf{l},\mathbf{q}) B'_{m,m'}(\ldots,\mathbf{l}) , \qquad (88)$$

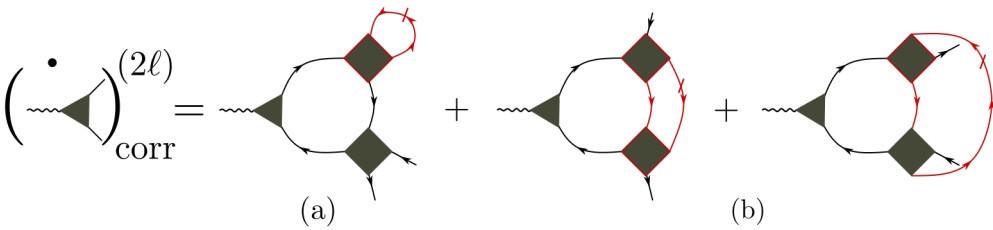

Figure 19: Simplified diagrammatic representation of the $2\ell$ correcting diagrams for the flow equation of $\gamma_3^\Lambda$ illustrating the topological structure of the diagrams. Diagram (a) can be reabsorbed in the single-scale propagator according to the "Katanin correction", while the second and the third contributions (b) represent the so-called "overlapping-diagrams".

where ... stands for the channel specific translation of the frequency dependencies.

We exemplify the projection for the channel $D$ to $P$. In momentum space, it reads

$$[\hat{P}[\phi_{ph}](i\omega_l, i\nu_o, i\nu_{o'}, \mathbf{q})]_{n,n'} = \int d\mathbf{k}d\mathbf{k}' f_n^*(\mathbf{k}) f_{n'}(\mathbf{k}') \times$$

$$\phi_{ph}\Big(i\omega_l - i\nu_{o'}, i\nu_o, i\nu_{o'}, \mathbf{q} - \mathbf{k}' - \mathbf{k}, \mathbf{k}, \mathbf{k}'\Big)$$

$$= \sum_{m,m'} \int d\mathbf{k}d\mathbf{k}' f_n^*(\mathbf{k}) f_{n'}(\mathbf{k}') f_m(\mathbf{k}) f_{m'}^*(\mathbf{k}') \times$$

$$D_{m,m'}(i\omega_l - i\nu_{o'}, i\nu_o, i\nu_{o'}, \mathbf{q} - \mathbf{k}' - \mathbf{k}). \tag{89}$$

We now transform the form factors to real space and shift the momentum dependence in order to get the matrix form of (88)

$$[\hat{P}[\phi_{ph}](i\omega_l, i\nu_o, i\nu_{o'}, \mathbf{q})]_{n,n'} = \sum_{m,m'} \int_{-\pi}^{\pi} d\mathbf{K} \sum_{\mathbf{R}\mathbf{R_1}\mathbf{R_2}} e^{i\mathbf{l}\mathbf{R} - i\mathbf{q}\mathbf{R}} f_n^*(\mathbf{R_1} - \mathbf{R}) f_{n'}(\mathbf{R_2} + \mathbf{R}) \times$$

$$f_m(\mathbf{R_1}) f_{m'}^*(\mathbf{R_2}) D_{m,m'}\Big(i\omega_l - i\nu_{o'} - i\nu_o, i\nu_o, i\nu_{o'}, \mathbf{l}\Big). \tag{90}$$

The same procedure for every channel projection leads to the matrix equations

$$[\hat{P}[\phi_{ph}](i\omega_l, i\nu_o, i\nu_{o'}, \mathbf{q})]_{n,n'} = \sum_{m,m',\mathbf{l}} A_{n,n',m,m'}^{P,D}(\mathbf{l}, \mathbf{q}) D_{m,m'}\Big(i\omega_l - i\nu_{o'} - i\nu_o, i\nu_o, i\nu_{o'}, \mathbf{l}\Big) \tag{91a}$$

$$[\hat{P}[\phi_{\overline{ph}}](i\omega_l, i\nu_o, i\nu_{o'}, \mathbf{q})]_{n,n'} = \sum_{m,m',\mathbf{l}} A_{n,n',m,m'}^{P,C}(\mathbf{l}, \mathbf{q}) C_{m,m'}\Big(-i\nu_o + i\nu_{o'}, i\nu_o, i\omega_l - i\nu_{o'}, \mathbf{l}\Big) \tag{91b}$$

$$[\hat{D}[\phi_{pp}](i\omega_l, i\nu_o, i\nu_{o'}, \mathbf{q})]_{n,n'} = \sum_{m,m',\mathbf{l}} A_{n,n',m,m'}^{D,P}(\mathbf{l}, \mathbf{q}) P_{m,m'}\Big(i\omega_l + i\nu_o + i\nu_{o'}, i\nu_o, i\nu_{o'}, \mathbf{l}\Big) \tag{91c}$$

$$[\hat{D}[\phi_{\overline{ph}}](i\omega_l, i\nu_o, i\nu_{o'}, \mathbf{q})]_{n,n'} = \sum_{m,m',\mathbf{l}} A_{n,n',m,m'}^{D,C}(\mathbf{l}, \mathbf{q}) C_{m,m'}\Big(i\nu_{o'} - i\nu_o, i\nu_o, i\nu_o + i\omega_l, \mathbf{l}\Big) \tag{91d}$$

$$[\hat{C}[\phi_{pp}](i\omega_l, i\nu_o, i\nu_{o'}, \mathbf{q})]_{n,n'} = \sum_{m,m',\mathbf{l}} A_{n,n',m,m'}^{C,P}(\mathbf{l}, \mathbf{q}) P_{m,m'}\Big(i\omega_l + i\nu_o + i\nu_{o'}, i\nu_o, i\omega_l + i\nu_o, \mathbf{l}\Big) \tag{91e}$$

$$[\hat{C}[\phi_{ph}](i\omega_l, i\nu_o, i\nu_{o'}, \mathbf{q})]_{n,n'} = \sum_{m,m',\mathbf{l}} A_{n,n',m,m'}^{C,D}(\mathbf{l}, \mathbf{q}) D_{m,m'}\Big(i\nu_{o'} - i\nu_o, i\nu_o, i\nu_o + i\omega_l, \mathbf{l}\Big), \tag{91f}$$

with the following projection matrices for the non-symmetrized notation

$$A_{n,n',m,m'}^{P,D}(\mathbf{l}, \mathbf{q}) = \sum_{\mathbf{R}\mathbf{R_1}\mathbf{R_2}} e^{i\mathbf{l}\mathbf{R} - i\mathbf{q}\mathbf{R}} f_n^*(\mathbf{R_1} - \mathbf{R}) f_{n'}(\mathbf{R_2} + \mathbf{R}) f_m(\mathbf{R_1}) f_{m'}^*(\mathbf{R_2}) \tag{92a}$$

$$A_{n,n',m,m'}^{P,C}(\mathbf{l}, \mathbf{q}) = \sum_{\mathbf{R}\mathbf{R_1}\mathbf{R_2}} e^{i\mathbf{l}\mathbf{R} + i\mathbf{q}\mathbf{R_2}} f_n^*(\mathbf{R_1} - \mathbf{R}) f_{n'}(-\mathbf{R_2} - \mathbf{R}) f_m(\mathbf{R_1}) f_{m'}^*(\mathbf{R_2}) \tag{92b}$$

$$A_{n,n',m,m'}^{D,P}(\mathbf{l}, \mathbf{q}) = \sum_{\mathbf{R}\mathbf{R_1}\mathbf{R_2}} e^{i\mathbf{l}\mathbf{R} - i\mathbf{q}\mathbf{R}} f_n^*(\mathbf{R_1} + \mathbf{R}) f_{n'}(\mathbf{R_2} - \mathbf{R}) f_m(\mathbf{R_1}) f_{m'}^*(\mathbf{R_2}) \tag{92c}$$

$$A_{n,n',m,m'}^{D,C}(\mathbf{l}, \mathbf{q}) = \sum_{\mathbf{R}\mathbf{R_1}\mathbf{R_2}} e^{i\mathbf{l}\mathbf{R} + i\mathbf{q}\mathbf{R_2}} f_n^*(\mathbf{R_1} - \mathbf{R_2} - \mathbf{R}) f_{n'}(-\mathbf{R}) f_m(\mathbf{R_1}) f_{m'}^*(\mathbf{R_2}) \tag{92d}$$

$$A_{n,n',m,m'}^{C,P}(\mathbf{l}, \mathbf{q}) = \sum_{\mathbf{R}\mathbf{R_1}\mathbf{R_2}} e^{i\mathbf{l}(\mathbf{R_2} - \mathbf{R}) + i\mathbf{q}\mathbf{R}} f_n^*(\mathbf{R_1} - \mathbf{R}) f_{n'}(\mathbf{R} - \mathbf{R_2}) f_m(\mathbf{R_1}) f_{m'}^*(\mathbf{R_2}) \tag{92e}$$

$$A_{n,n',m,m'}^{C,D}(\mathbf{l}, \mathbf{q}) = \sum_{\mathbf{R}\mathbf{R_1}\mathbf{R_2}} e^{i\mathbf{l}\mathbf{R} + i\mathbf{q}\mathbf{R_2}} f_n^*(\mathbf{R_1} - \mathbf{R_2} - \mathbf{R}) f_{n'}(-\mathbf{R}) f_m(\mathbf{R_1}) f_{m'}^*(\mathbf{R_2}). \tag{92f}$$

## G   Performance of the code

The results shown in this paper were obtained with an OpenMP parallelized code on a single node. In Fig. 3, the scaling in memory and calculation effort is illustrated. The use of symmetries can decrease the calculation effort considerably. The maximum computing time using 40 threads was obtained for the following set of parameters (see caption of Fig. 3)

$$N_\nu = 8 \quad N_\mathbf{q} = 256 \quad N_{FF} = 1 \quad T = 0.125 \quad \ell = 8,$$

giving $\tau_{\max} = $ total CPU time$/(40\text{CPUs}) \sim 10$ days. The memory usage of a process for this set of parameters is approximately 15 GiB.

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
