# Peer review of "Multiloop functional renormalization group for the two-dimensional Hubbard model: Loop convergence of the response functions"

_SciPost Physics, doi:SciPost Phys. 6, 009 (2019)_

## Round 2 · Referee Report · Anonymous (Referee 1) · 2018-9-10

Strengths

(1) Manuscript develops and discusses a very advanced many-body method
(2) This method is applied to a complicated model (the 2D Hubbard model)
(3) Many convergence and benchmark checks are performed
(4) The developed method has much potential for future applications

Weaknesses

(1) Some technical aspects could be better illustrated (see below)
(2) The violation of the Mermin-Wagner theorem could be better discussed
(3) Possible qualitatively new results for the 2D Hubbard model could be better highlighted (see below)

Report

This manuscript is about the development and application of a fRG algorithm for two-dimensional Hubbard models, improving previous implementations in various different aspects. Firstly, the full frequency and momentum dependence of the vertex functions are explicitly accounted for and, secondly, the feedback of the self energy into the two-particle vertex flow is included. Furthermore, the multi-loop fRG scheme is implemented which successively adds higher loop orders to the diagrammatic approximation and allows to systematically improve the scheme. The implementation of the multi-loop fRG also represents the main advancement compared to previous fRG studies of two-dimensional Hubbard models. Another focus lies on response functions which are computed in two different ways, via a post-processing of the vertex functions and via explicitly solving the flow equations for the susceptibilities. In the latter case, the authors show how these flow equations can be generalized to a multi-loop structure. As a test system, the authors apply their technique to the two dimensional Hubbard model on the square lattice at half filling. Extensive convergence and benchmark checks concerning the number of fermionic frequencies and momentum patching points, the impact of the self-energy insertion, and the number of loops are performed. As a main result, the authors find a rapid convergence of the susceptibility when increasing the number of loops. This convergence also applies to the aforementioned two methods of computing the susceptibility.

Overall, I find it quite impressive how this work improves the fRG on so many different fronts. Even though the paper has a clear methodological focus I am also impressed that the authors manage to apply their technique to a quite non-trivial system such as the two dimensional Hubbard model. Indeed, what they call their "test model" is already a very complex system. Finally, the comprehensive and careful convergence checks add a significant amount of trust in the validity of their analysis. I see significant potential of this method for future applications to other (possibly more complicated) Hubbard models. For these reasons I recommend the publication of this work after the authors have considered the points/questions listed below.

Requested changes

(1) The Section 2.2 about the derivation of the multi-loop fRG equations for the susceptibility is rather technical and I had problems understanding how this multi-loop extension works. The section would profit much from a diagrammatic illustration of the scheme. In Fig. 1 an illustration is already given, but I didn't find it very enlightening.

(2) As shown in Fig. 6 the real part of the self energy vanishes at two points at the Fermi surface; hence at these points the Fermi surface remains unchanged by the self energy feedback. Does this apply to the whole Fermi surface and for higher loop orders as well? In other words, does the perfect nesting effect always remain intact?

(3) The authors argue that in the limit of large loop numbers, the Mermin-Wagner theorem should be fulfilled. On the other hand, their numerical results in Fig. 10 still show a significant violation of Mermin-Wagner even at l=8. The authors provide arguments why it is very challenging to suppress the pseudo-critical temperature in the absence of extremely long-range fluctuations. However, given the fact that the fRG is often criticized because of the violation of Mermin-Wagner, it would be desirable to see a more detailed discussion here. What I mean is, if a method that is known to fulfill Mermin-Wagner still finds a finite ordering temperature, then this temperature should have a physical meaning. For example, it should be related to the extent of correlations taken into account and it should maybe even be possible to calculate/estimate it without solving the fRG equations. Can the authors comment on this?

(4) Besides quantitative changes in the results when applying their extended fRG scheme to the two-dimensional Hubbard model, do the authors also find qualitatively new properties of this system? If yes, I think it should be better highlighted.

(5) It would be very interesting to know about the numerical efforts of these calculations. How long did they take and on how many CPUs did they run? Do these calculations require massive parallelization and supercomputing facilities?

(6) The numbering of the subsections in Sec. 4 is very strange as it starts with a zeroth subsection.

---

## Round 3 · Referee Report · Anonymous · 2018-12-10

Strengths
(1) Manuscript develops and discusses a very advanced many-body method
(2) This method is applied to a complicated model (the 2D Hubbard model)
(3) Many convergence and benchmark checks are performed
(4) The developed method has much potential for future applications
Weaknesses
(1) The results do not fulfill the Mermin-Wagner theorem but the reasons for this failure are well discussed
Report
The resubmitted manuscript adequately addresses all suggestions and questions raised in my previous report. It features a more expanded discussion of various important aspects of this work. I remain to be impressed by the fact that this paper improves the fRG in so many different directions. I am, therefore, happy to recommend the manuscript for publication.
Sabine Andergassen on 2019-10-25 [id 633]
A few minor corrections on this paper, including some typos in formulas, are included in un updated arXiv:1807.02697 version.

---

## Round 3 · Author Response

\vskip 5mm
\noindent
We thank the Referee for carefully reading our manuscript and for posing constructive questions and suggestions to improve on the readibility of our work. We are glad that she/he finds {\sl ``[...] it quite impressive how the work improves in the fRG on so many fronts.''} and sees {``\sl [...] significant potential of this method for future applications to other (possibly more complicated) Hubbard models''}. Below we provide the clarification of all specific questions/remarks.
\vskip 5mm
\noindent
{\it 1. ``The Section 2.2 about the derivation of the multi-loop fRG equations for the susceptibility is rather technical and I had problems understanding how this multi-loop extension works. The section would profit much from a diagrammatic illustration of the scheme. In Fig. 1 an illustration is already given, but I didn't find it very enlightening."}\vspace{2mm}\\
\noindent
To this purpose, following the useful suggestion of the Referee, we added Fig.~1 in the revised version of the manuscript. The latter aims at giving a diagrammatic illustration how to construct the multiloop corrections upon the standard one-loop truncated flow equations of the susceptibility and the fermion-boson vertex $\gamma_3$ in all physical channels. Similarly to the illustration of the multiloop flow equations for the channel-decomposed vertices in [Kugler Phys. Rev. B {\bf 97} (2018), Fig.~5], we proceed by showing the two-loop correction-terms (Fig.~1(b) in our revised version of the manuscript) and by generalizing the corrections to an arbitrary $2+\ell \geq 3$-loop order (Fig.~1.(c)). We hope that Fig.~2 (Fig.~1 in the old version of the manuscript) can now be more clearly interpreted as an example of the lowest multiloop corrections, up to $\ell \!=\! 3$ and $\ell \!=\!2$ for the flow equation of the susceptibility and fermion-boson vertex, respectively, at the leading order in the (bare) interaction.
\vskip 5mm
\noindent
{\it 2. ``As shown in Fig. 6 the real part of the self energy vanishes at two points at the Fermi surface; hence at these points the Fermi surface remains unchanged by the self energy feedback. Does this apply to the whole Fermi surface and for higher loop orders as well? In other words, does the perfect nesting effect always remain intact?''}\vspace{2mm}\\
The vanishing of the Fermi surface shift is caused by the particle-hole symmetry of the model at half filling. In this situation, one can easily show that the self-energy fulfill the symmetry property which is reported in the revised version of the manuscript as Eq.~53. In the text we added additional comments regarding the consequence of the particle-hole symmetry on the structure of the self-energy, and hence on the spectral properties of the system at the highly symmetric points of the Brillouine Zone, which have been shown in Fig.~7 (Fig.~6 in the old version of the manuscript). As specified in the text {\it `` This symmetry is not violated by any of the perturbative corrections and also not by the numerical implementation (e.g. the choice of k-points in the BZ).."}. The confirmation that the particle-hole symmetry of the self-energy is preserved at higher loop-order has been numerically displayed in Fig.~7 (Fig.~6 in the old version of the manuscript) where we added the $\ell=8$ mfRG results for the real part (left panel) and imaginary part (right panel) of the self-energy.\\ \noindent Interestingly, the higher loop-order results do not show any appreciable difference in the bandwidth renormalization with respect to the corresponding $\ell=1$ ones, and the multiloop corrections provide only a slight reduction of the scattering rate (more enhanced at the FS) for all ${\bf k}$-points considered.
\vskip 5mm
\noindent
{\it 3. ``The authors argue that in the limit of large loop numbers, the Mermin-Wagner theorem should be fulfilled. On the other hand, their numerical results in Fig. 10 still show a significant violation of Mermin-Wagner even at l=8. The authors provide arguments why it is very challenging to suppress the pseudo-critical temperature in the absence of extremely long-range fluctuations. However, given the fact that the fRG is often criticized because of the violation of Mermin-Wagner, it would be desirable to see a more detailed discussion here. What I mean is, if a method that is known to fulfill Mermin-Wagner still finds a finite ordering temperature, then this temperature should have a physical meaning. For example, it should be related to the extent of correlations taken into account and it should maybe even be possible to calculate/estimate it without solving the fRG equations. Can the authors comment on this?''}\vspace{2mm}\\
\noindent
We thank the Referee for the comment and we fully agree that this point deserves a more precise clarification (which we have also included, more shortly, in the revised version of our manuscript).
\noindent
The fulfillment of the Mermin-Wagner (MW) Theorem in mfRG is formally guaranteed by the fact that for $\ell \rightarrow \infty$, one reproduces the PA per construction (which was demonstrated, in turn, to guarantee the MW theorem). Nonetheless, the extraction of the pseudocritical temperature from a set of a numerical mfRG data (e.g, our $\ell=8$ data of Fig.~11) is highly non trivial, because of other issues.
\noindent
One is physical, as the fulfillment of the MW theorem implies the emergence, at a very low-T scale (let's call it here : $T^{*}$), of antiferromagnetic (AF) fluctuations, growing exponentially (and no longer with a power law) for $T < T^{*}$. This is also reflected into the associated AF (inverse) susceptibility which will display an high-T linear behavior for $T \gg T^{*}$ (where the spatial fluctuations are short range and an effective mean-field might work reasonably), which is eventually reverted in an exponential behavior for $T \ll T^{*}$.
\noindent The second reason is computational, as the convergence of the mfRG calculations w.r.t. to the momentum and frequency-grid becomes particularly though in the aforementioned low-T domain of quasi-long range fluctuations. The ability to correctly capture the latter is essential to obtain the suppression of $T^{*}$ down to exactly $T\!=\!0$.
\noindent
On the basis of the above considerations, the interpretation of our multiloop mfRG data becomes rather clear. In the region where our higher loop calculations could be converged w.r.t. the underlying momentum/frequency-grids, we have also obtained loop-converged results (namely, $\ell=8$) to the PA. This indicates a systematic suppression of $\chi_{\text{AF}}$ w.r.t. the corresponding $\ell$-loop results, due to the additional fluctuations included, which is a trend in perfect accordance with the theoretical expectations based on the MW theorem.
At the same time, we note that the temperatures considered are certainly larger than $T^{*}$, as our (loop-converged, PA) data for $\chi_{\text{AF}}^{-1}(T)$ display a linear behavior.
Within the present implementation, we could not get, instead, momentum-converged $8$-loop results in the (quasi-long range fluctuations regime) of $T < T^{*}$ of the $U=2t$ half-filled case. Hence, it is simply not possible to extrapolate a corresponding $T_{\text{pc}}$ from our mfRG converged data, because they are all lying in the ``effective mean-field" regime. Such an extrapolation could be rather regarded as the instability of a renormalized mean-field theory (such as, e.g., the DMFT), which yields typically much lower instability temperatures than the standard (static) mean-field. This interpretation is fully supported by quantitative estimates in our case. In fact, by linearly extrapolating our $8$-loop results, we would get a temperature $\sim 0.07 t$, which is much lower than the corresponding RPA estimates of $\sim 0.21 t$ [C. Eckhardt et al., PRB 2018)] but quantitatively in agreement with the DMFT results [see Sec. IV in RMP 90 025003 (2018)]. We note that, physically, this T-scale would be roughly consistent with the T interval where a sharp crossover from a paramagnetic metallic to a paramagnetic insulating state is found, and, hence, at weak-coupling, also with the aforementioned $T^{*}$-scale.
The latter interpretation finds further support in the numerical results of diagrammatic extensions of DMFT applied to the low-T regime of the 2d Hubbard model [see: T. Sch\"afer et al, PRB 2015, and Sec. IV in RMP 90 025003 (2018)].
\vskip 5mm
\noindent
{\it 4. ``Besides quantitative changes in the results when applying their extended fRG scheme to the two-dimensional Hubbard model, do the authors also find qualitatively new properties of this system? If yes, I think it should be better highlighted. ''}\vspace{2mm}\\
The 2D Hubbard model at half-filling represents an highly investigated system whose underlying features have been studied by means of a huge variety of methods (see, e.g., the extensive review by J. P. F. LeBlanc et. al. [Phys. Rev. X (2015)]).
At the present stage, the description of physical properties computed by means of the different approaches appears to match qualitatively, and in significant parameter region also quantitatively, if calculations are performed in the particle-hole symmetric sector of this model, where the physics is controlled essentially by antiferromagnetic correlations.
The 2D Hubbard model at half-filling represents thus a rather solid and, at the same time, physically not-trivial situation to verify the applicability of our multiloop-extended fRG scheme.
\noindent The fRG method itself is, however, not restricted to the particle-hole symmetric sector, or to systems dominated by a specific type of correlations. On the contrary, one of the strength of the fRG, proven by the several 1-loop studies in literature [cf. W. Metzner et al, RMP (2012)], is the possibility to analyze situations where the landscape of instabilities and emergent energy scales is not clearly settled a priori. To be more specific, we recall that, over the last decades, fRG has been extensively applied in more general situations, with a considerable number of studies of the 2D Hubbard model out-of-half filling. These studies were performed, however, with a lower quantitative control w.r.t. our presented algorithm (see Sec.~3), which motivates the effort of the extensions presented in our work.
\noindent Hence, summarizing the considerations above, because of the the strong leading antiferromagnetic fluctuations dominating the half-filling/particle-hole symmetric case, the high-loop corrections can not (and do not) lead to any qualitative change of the results found for $\ell=1$, {\sl but} for the fulfillment of the Mermin-Wagner theorem, i.e., for the disappearance of any pseudocritical (magnetic) divergence wherever the mfRG calculation can be fully converged. The discussion of this point has been refined in the revised manuscript (see reply to the previous point). The highly studied sector of half-filling represents, in any case, the most natural test-bed to benchmark and present the fundamental performance of our improved approach.
\noindent On a longer time perspective, non-particle-hole symmetric situations offer a very promising ``play-ground" for our newly developed mfRG algorithm. The situation in which fluctuations of different nature become simultaneously large represents an evident case, where multiloop corrections could play an crucial role in modifying, qualitatively, the phase diagram of the system. In fact, as discussed in the manuscript, one of the most important effect of the multiloop corrections is to change the balance between the strength of the different fluctuating channels. Hence, in the proximity of competing instabilities, such changes might lead to radically different outcomes w.r.t. some previous fRG studies [C. J. Halboth and W. Metzner, Phys. Rev. B {\bf 61} (2000), D. Zanchi and H. J. Schulz Phys. Rev. B {\bf 61},C. J. Halboth and W. Metzner, Phys. Rev. Lett. {\bf 24} (2000), C. Husemann and M. Salmhofer Phys. Rev. B {\bf 79} (2009), K.-W. Giering and M. Salmhofer Phys. Rev. B {\bf 86} (2012)].
\noindent In the resubmitted version, we have amended the conclusion section with a similar statement to make this point clearer.
\vskip 5mm
\noindent
{\it 5. `It would be very interesting to know about the numerical efforts of these calculations. How long did they take and on how many CPUs did they run? Do these calculations require massive parallelization and supercomputing facilities?''}\vspace{2mm}\\
\noindent
To this purpose we have added an additional subsection, namely Sec.3.3 {\it "Performance of the code"}, which should provide the reader information regarding the multithreading parallelization of the code and a rough estimate of the computing time needed for the longest calulation perfomed in the paper. The scaling with respect to a different parameter choice can be deduced by looking at Fig.~3 (Fig.~2 in the old version of the manuscript).
\vskip 5mm
\noindent
{\it 6. `The numbering of the subsections in Sec. 4 is very strange as it starts with a zeroth subsection.''}\vspace{2mm}\\
\noindent
We thank the Referee for noticing this error which has occured in the formatting process. In the revised version of our manuscript, the enumeration of Sec.4 has been corrected.

---

## Round 3 · List of Changes

{\bf LIST OF CHANGES}
\vskip 5mm
\begin{enumerate}[label=\roman{*}), ref=(\roman{*})]
\item We have added the new Fig.~1 which illustrates the diagrammatic representation of the multiloop flow equation for the susceptibility and the fermion-boson vertex.
\item We have added to Fig.~7 (left and right) mfRG results for the self-energy corresponding to $\ell=8$ .
\item We have added a short discussion regarding the properties of the self-energy at paticle-hole symmetry referring to the $1\ell$ results (see Sec.~4.4) and to the mfRG results (see Sec. 4.5).
\item We have added a short sentence in Sec.~4.5 p.~26-27 \textcolor{red}{@Ale:todo?}.
\item We have corrected the numbering of the subsections in Sec.~4 according to the remark (6) from the Referee.
\item We have added some sentences in the Conclusion, to answer question (5) from the Referee.
\item Funding information? Cluster at MPI.
\item We added Figure 14 (Appendix A) for a better understanding of the frequency and
momentum notations used.
\item We have corrected a typo in Eq.~53a)-c) in Appendix A and in Eq.~92e) in Appendix F.
\item We have added Appendix G {\it ``Performance of the code"}.
\item We have corrected typos throughout the manuscript.
\end{enumerate}

---

## Editorial Decision

published